

# Impacts of Household Sources on Air Pollution at Village and Regional Scales in India

Brigitte Rooney[1], Ran Zhao[1,2], Kelvin H. Bates[1,3], Ajay Pillarisetti[4], Sumit Sharma[5], Seema Kundu[5], Tami C. Bond[6], Nicholas L. Lam[6,7], Bora Ozaltun[6], Li Xu[6], Lauren T. Fleming[8], Robert Weltman[8], Simone Meinardi[8], Donald R. Blake[8], Sergey A. Nizkorodov[8], Rufus D. Edwards[9], Ankit Yadav[10], Narendra K. Arora[10], Kirk R. Smith[4], and John H. Seinfeld[1]

[1]Division of Chemistry and Chemical Engineering and Division of Engineering and Applied Science, California Institute of Technology, Pasadena, CA 91125, USA

[2]Current address: Department of Chemistry, University of Alberta, Edmonton, Alberta, Canada T6G 2R3

[3]Current address: Center for the Environment, Harvard University, Cambridge, MA 02138, USA

[4]School of Public Health, University of California, Berkeley, CA 94720, USA

[5]The Energy and Resources Institute (TERI), New Delhi-110003, India

[6]Department of Civil and Environmental Engineering, University of Illinois, Urbana-Champaign, IL 61801, USA

[7]Current address: Schatz Energy Research Center, Humboldt State University, Arcata, CA 95521, USA

[8]Department of Chemistry, University of California, Irvine CA 92697, USA

[9]Department of Epidemiology, University of California, Irvine, CA 92697, USA

[10]The INCLEN Trust, Okhla Industrial Area, Phase-I, New Delhi-110020, India

**Correspondence:** John H. Seinfeld (seinfeld@caltech.edu) and Kirk R. Smith (krksmith@berkeley.edu)

## Abstract

Approximately 3 billion people worldwide cook with solid fuels, such as wood, charcoal, and agricultural residues. These fuels are often combusted in inefficient cookstoves, producing carbonaceous emissions. Between 2.6 and 3.8 million premature deaths occur as a result to





exposure to fine particulate matter from the resulting household air pollution (Health Effects Institute, 2018a; World Health Organization, 2018). Household air pollution also contributes to ambient air pollution; the magnitude of this contribution is uncertain. Here, we simulate the distribution of the two major health-damaging outdoor air pollution species ($PM_{2.5}$ and $O_3$) using

state-of-the-science emissions databases and atmospheric chemical transport models to estimate the impact of household combustion on ambient air quality in India. The present study focuses on New Delhi and the SOMAARTH Demographic, Development, and Environmental Surveillance Site (DDESS) in the Palwal District of Haryana, located about 80 km south of New Delhi. The DDESS covers an approximate population of 200,000 within 52 villages. The emissions

inventory used in the present study was prepared based on a national inventory in India (Sharma et al., 2015, 2016), an updated residential sector inventory prepared at the University of Illinois, updated cookstove emissions factors from Fleming et al. (2018b), and $PM_{2.5}$ speciation from cooking fires from Jayarathne et al. (2018). Simulation of regional air quality was carried out using the U.S. Environmental Protection Agency Community Multiscale Air Quality modeling system

(CMAQ), in conjunction with the Weather Research and Forecasting modeling system (WRF) to simulate the meteorological inputs for CMAQ, and the global chemical transport model GEOS-Chem to generate concentrations on the boundary of the computational domain. Comparisons between observed and simulated $O_3$ and $PM_{2.5}$ levels are carried out to assess overall airborne levels and to estimate the contribution of household cooking emissions. Observed and predicted

ozone levels over New Delhi during September 2015, December 2015, and September 2016 routinely exceeded 150 µg m$^{-3}$, as compared with the 8-hour Indian standard of 100 µg m$^{-3}$, and, on occasion, exceeded 200 µg m$^{-3}$. $PM_{2.5}$ levels are predicted over the SOMAARTH headquarters (September 2015 and September 2016), Bajada Pahari (a village in the surveillance site, September 2015, December 2015, and September 2016), and New Delhi (September 2015,

December 2015, and September 2016). Predicted levels vary depending on the time of year but, on the whole, tend to be somewhat less than those observed. The predicted fractional impact of residential emissions on $PM_{2.5}$ levels varies from about 0.30 in SOMAARTH HQ and Bajada Pahari to about 0.10 in New Delhi. Predicted levels of secondary organic $PM_{2.5}$ during the periods studied at the three locations averaged about 5 µg m$^{-3}$, representing approximately 10% of total





PM$_{2.5}$ levels, accentuating the dominant role played by primary carbonaceous emissions in all three locations.

**5    Introduction**

Although outdoor air pollution is widely recognized as a health risk, quantitative understanding remains uncertain on the degree to which household combustion contributes to unhealthy air. Recent studies in China, for example, show that 50-70% of black carbon emissions and 60-90% of organic carbon (OC) emissions can be attributed to residential coal and biomass burning (Cao et

al., 2006; Klimont et al., 2009; Lai et al., 2011). Moreover, existing global emissions inventories show a significant contribution of household sources to primary PM$_{2.5}$ (particulate matter of diameter less than or equal to 2.5 micrometers) emissions. The Indo-Gangetic Plain of Northern India (23-31$^{o}$N, 68-90$^{o}$E) has among the world's highest values of PM$_{2.5}$. In this region, the major sources of emissions of primary PM$_{2.5}$ and of precursors to secondary PM$_{2.5}$ are coal-fired power

plants, industries, agricultural biomass burning, transportation, and combustion of biomass fuels for heating and cooking (Reddy and Venkataraman, 2002; Rehman et al., 2011). The southwest monsoon in summer months in India leads to lower pollution levels than in winter months, which are characterized by low wind speeds, shallow boundary layer depths, and high relative humidity (Sen et al., 2017).  With the difficulty in determining representative emissions estimates (Jena et

al., 2015; Zhong et al., 2016), simulating the extremely high PM$_{2.5}$ observations in the Indo-Gangetic Plain has remained a challenge (Schnell et al., 2018).

Approximately 3 billion people worldwide cook with solid fuels, such as wood, charcoal, and agricultural residues (Bonjour et al., 2013; Chafe et al., 2014; Smith et al., 2014; Edwards et

al., 2017). Such solid fuels are often combusted in inefficient cookstoves, producing black carbon (BC) and organic carbon emissions. Between 2.6 and 3.8 million premature deaths occur as a result to exposure to fine particulate matter from household air pollution (Health Effects Institute, 2018a; World Health Organization, 2018). In India, more than 50% of households report use of wood or crop residues, and 8% report use of dung as cooking fuel (Klimont et al., 2009;



Census of India, 2011; Pant and Harrison, 2012). Residential biomass burning is one of the largest individual contributors to the burden of disease in India, estimated to be responsible for 780,000 premature deaths in 2016 (Indian Council of Medical Research et al., 2017). The recent GBD MAPS Working Group (Health Effects Institute, 2018b) estimated that household emissions in

5 India produce about 24% of ambient air pollution exposure. Coal combustion, roughly evenly divided between industrial sources and thermal power plants, was estimated by this study to be responsible for 15.3% of exposure in 2015. Open burning of agricultural crop stubble was estimated annually to be responsible for 6.1% nationally, although more important in some areas.

Traditional biomass cookstoves, with characteristic low combustion efficiencies, produce significant gas- and particle-phase emissions. An early study of household air pollution in India found outdoor total suspended particulate matter (TSP) levels in four Gujarati villages well over 2 mg m$^{-3}$ during cooking periods (Smith et al., 1983). Secondary organic aerosol, produced by gas-

15 phase conversion of volatile organic compounds to the particulate phase, is also important in ambient PM levels, yet no prediction for secondary organic aerosols in India exists. Overall, household cooking in India has been estimated by various groups to produce 22-50% of ambient PM$_{2.5}$ exposure (Butt et al., 2016; Chafe et al., 2014; Conibear et al., 2018; Health Effects Institute, 2018b; Lelieveld et al., 2015; Silva et al., 2016), and Fleming et al. (2018a,b) report

20 characterization of a wide range of particle-phase compounds emitted by cookstoves. In a multi-model evaluation, Pan et al. (2015) concluded that an underestimation of biomass combustion emissions, especially in winter, was the dominant source of model underestimation. Here, we address both primary and secondary organic aerosols from household burning of biomass for cooking.

Air quality in urban areas in India is determined largely, but not entirely, by anthropogenic fuel combustion. In rural areas, residential combustion of biomass for household uses, such as cooking, also contributes to non-methane volatile organic carbon (NMVOC) and particulate emissions (Sharma et al., 2015, 2018). Average daily PM$_{2.5}$ levels frequently exceed the 24-hour



Indian standard of 60 µg m$^{-3}$ and can exceed 200 µg m$^{-3}$, even in rural areas. The local region on which the present study focuses is the SOMAARTH Demographic, Development, and Environmental Surveillance Site (DDESS) run by the International Clinical Epidemiological Network (INCLEN) in the Palwal District of Haryana (Figure 1). Located about 80 km south of New

Delhi, SOMAARTH covers an approximate population of 200,000 in 52 villages. Particular focus is given to the SOMAARTH Headquarters (HQ) and the village of Bajada Pahari within DDESS, coinciding with the work of Fleming et al. (2018b), who studied cookstove non-methane hydrocarbon (NMHC) emissions and ambient air quality. Demographically, with a coverage of almost 308 sq km, the DDESS has a mix of populations from different religions and socioeconomic

and development statuses.

The climate of the region of interest in the present study is primarily influenced by monsoons, with a dry winter and very wet summer. The rainy season, July through September, is characterized by average temperatures around 30 °C and primarily easterly and southeasterly

winds. In a study related to the present one, Schnell et al. (2018) used emission datasets developed for the Coupled Model Intercomparison Project Phases 5 (CMIP5) and 6 (CMIP6) to evaluate the impact on predicted PM$_{2.5}$ over Northern India, October-March 2015-2016, with special attention to the effect of meteorology of the region, including relative humidity, boundary layer depth, strength of the temperature inversion, and low level wind speed. In that work,

nitrate and organic matter (OM) were predicted to be the dominant components of total PM$_{2.5}$ over most of Northern India.

The goal of the present work is twofold: (1) Simulate the distribution of primary and secondary PM$_{2.5}$ and O$_3$ production using recently updated emissions databases and atmospheric

chemical transport models to obtain estimates of the total impact on ambient air quality attributable to household combustion; and (2) Assess the fraction of the two major health-damaging outdoor air pollution species (PM$_{2.5}$ and O$_3$) that can be attributed to household combustion in India. With respect to ozone, the present work follows that of Sharma et al. (2016) who simulated regional and urban ozone concentrations in India using a chemical transport



model and included a sensitivity analysis to highlight the effect of changing precursor species on $O_3$ levels. The present work is based on simulating the levels of both $O_3$ and $PM_{2.5}$ at the regional level based on recent emissions inventories using state-of-the-science atmospheric chemical transport models.

### Emissions Inventory

The TERI national inventory for India (Sharma et al., 2015, 2016) emissions grid was updated with high resolution estimates of residential sector emissions. TERI includes source-sector categories

10 for transportation, industry, power, waste management, agriculture, oil and gas, diesel power generation, and residential energy use at a native spatial resolution of 36 km. It is compiled on a yearly basis with monthly variations for brick kilns and agricultural burning, and diurnal variation for the residential sector. Thus, daily emission rates are generated for all species and sectors, except for the residential sector. All emissions are assumed to occur at Earth's surface.

  To examine local and regional impacts of residential sector emissions in greater detail, an update to the TERI inventory was performed to consider more granular input data specific to the residential sector. Bottom-up estimates of delivered energy for cooking, space heating, water heating, and lighting were informed by those used in Pandey et al. (2014), and converted to fuel

20 consumption at the village level using population size and percentage of reported primary cooking and lighting fuels from the 2011 Census of India (Census of India, 2011). Urban areas of the domain were assumed to have the average cooking and lighting fuel use profiles of the average urban areas of their district. Fuel consumption was converted to emission rates using fuel-specific emission factors informed by a review of field and laboratory studies, which was

25 used to update the Speciated Pollutant Emissions Wizard (SPEW) inventory (Bond et al., 2004) and to generate summary estimates by fuel type. Hourly emissions were generated using source-specific diurnal emissions profiles (Figure 2). The same diurnal emissions profile is applied to all pollutant species from a source category and were informed by real-time emissions measurements taken in homes during cooking as part of Fleming et al. (2018a,b). Profiles for fuel-





based lighting were informed by real-time measurements of kerosene lamp usage data reported in Lam et al. (2018). The residential sector inventory represents surface emissions with a native spatial resolution of 30-arc seconds (~1 km).

In deriving summary estimates of emission factors, priority was given to emission factor measurements from field-based studies. Several studies have shown that laboratory-based measurements of stove and lighting emissions are lower than those of devices measured in actual homes (Roden et al., 2009), perhaps due to higher variation in fuel quality and operator behavior. Field-based emission factors utilized in this study included those for total non-methane

hydrocarbons, measured from fuels and stoves within our actual study domain (Fleming et al. 2018a,b). $PM_{2.5}$ speciation from cooking fires was informed by Jayarathne et al. (2018) (Tables 1-3). Residential emission rates for $PM_{2.5}$, black carbon (BC), organic carbon (OC), CO, $NO_x$, $CH_4$, $CO_2$, and total non-methane hydrocarbons (NMHC) were generated from SPEW. As our scenario focuses on the month of August, there are no emissions from heating within the study domain

and no hydrocarbon emissions from solvents are considered.

We employed various methods to account for pollutant species not explicitly reported by SPEW. To speciate total non-methane hydrocarbons (NMHC), we employed HC species-specific emission factors (Fleming et al. 2018b), differentiated by fuel and stove type (i.e. traditional

stove, or *chulha*, with wood or dung, and simmering stove, or *angithi,* with dung). The NMHC emission profile of dung was assumed to be the average of measurements from chulha and angithi stoves. Gas-phase $SO_2$ and $NH_3$ emissions were informed by existing residential emissions in the TERI inventory (Sharma et al. 2015); NO and $NO_2$ were estimated from $NO_x$ emissions assuming a NO:$NO_2$ emission ratio of 10:1.

Particle-phase speciation of total $PM_{2.5}$ into Na, $NH_4$, K, Cl, $NO_3^-$, and $SO_4^{2-}$ was based on PM mass emissions from wood- and dung-fueled cooking fires as reported by Jayarathne et al. (2018), and primary cooking fuel type distribution data from the 2011 census. For simplicity, we assume that all emissions in each computational grid cell are produced by either wood or dung,



whichever represents the greater fraction of PM$_{2.5}$ emissions in that cell (Figure 3). We assume that the PM$_{2.5}$ speciation contribution by LPG is negligible, given its extremely low PM$_{2.5}$ emission rate (Shen et al., 2018). The profile for agricultural residue in this area is assumed to be similar to wood, and therefore in cells where agriculture residue dominates, the wood speciation profiles

are applied, although not the overall emission factors. Non-carbon organic particulate matter (PNCOM), coarse mode particulate matter (PMC), and particulate water (PH$_2$O) were assumed to be negligible. Emissions of remaining particle-phase species (i.e. Al, Ca, Fe, Mg, Mn, Si, and Ti) are also assumed to be negligible. We define unspeciated fine particulate matter (PM$_{othr}$), as the portion of total PM$_{2.5}$ unassigned to any other species: $PM_{othr} = PM_{2.5} - (P_{Na} + P_{NH_4} + P_K + $

$P_{Cl} + P_{NO_3} + P_{SO_4})$.

**Air Quality Data**

Gas-phase air quality data analyzed in the present study come from the Central Pollution Control

Board (CPCB) of the Ministry of Environment, Forest & Climate Change, Government of India at the Punjabi Bagh station of west New Delhi. Particle-phase data analyzed come from the SOMAARTH Demographic, Development, and Environmental Surveillance Site (Mukhopadhyay et al., 2012; Pillarisetti et al., 2014; Balakrishnan et al., 2015) managed by the International Clinical Epidemiological Network (INCLEN). Palwal District has a population of ~ 1 million over an

area of 1400 km$^2$. In this district, ~ 39% of households utilize wood burning as their primary cooking fuel, with dung (~25%) and crop residues (~7%) (Census of India, 2011). Specific sites studied are the SOMAARTH headquarters (HQ) in Aurangabad (15 km south of Palwal) and the village of Bajada Pahari (8 km northwest of SOMAARTH HQ). Ambient measurement sites are shown in Figure 1, and Table 4 details available data for each location.

**Atmospheric Modeling**

Simulation of regional air quality was carried out using the U.S. Environmental Protection Agency Community Multiscale Air Quality modeling system (CMAQ), version 5.2 (Appel et al., 2017; US



EPA, 2017). CMAQ is a three-dimensional chemical transport model (CTM) that predicts the dynamic concentrations of airborne species. CMAQ includes modules of radiative processes, aerosol microphysics, cloud processes, wet and dry deposition, and atmospheric transport. Required input to the model includes emissions inventories, initial and boundary conditions, and

meteorological fields (Figure 4). The domain-specific, gridded emissions inventory provides hourly-resolved total emission rates for each species (not differentiated by source) by cell, timestep, and vertical layer. Initial conditions (ICs) and boundary conditions (BCs) are necessary to define the atmospheric chemical concentrations in the domain at the first time step and at the domain edges, respectively. Simulations operating with nested domains require two groups of

initial conditions and boundary conditions. The present study uses the global chemical transport model   GEOS-Chem   v11-02c   (acmg.seas.harvard.edu/geos/index.html)   to   generate concentrations on the boundary of the computational domain and CMAQ to produce initial and boundary conditions for the inner parent domain and nested domain, respectively. Meteorological conditions (including temperature, relative humidity, wind speed and direction

and land use and terrain data) drive the atmospheric processes represented in CMAQ. The Weather Research and Forecasting modeling system (WRF) – Advanced Research WRF (WRF-ARW, version 3.7), was used to simulate the meteorological input for CMAQ (Skamarock et al., 2008).

To study the impact of household emissions on ambient air pollution, we simulated two emission scenarios each for three time periods and over two nested domains (Tables 5 and 6). In Table 5, a "Total" emission scenario represents the overall atmospheric environment by including emissions from all source-sectors in the inventory. A "Non-Residential" emission scenario represents zeroing-out or "turning-off" all household emissions. By considering these scenarios

independently and comparing their output, we can isolate the effect of the residential sector on the ambient atmosphere.

Each scenario was simulated over a region in northern India with nested domains (Figure 1). This setup allows the nested domain to be simulated at a higher resolution and to use the



results of the parent CMAQ simulation to improve the boundary conditions. Figure 1 shows the "parent" domain which covers a 600 km by 600 km area including Delhi and portions of surrounding states at 4 km grid resolution. The finer "child" domain at 1 km grid resolution covers a 100 km x 100 km area south of New Delhi. Palwal District and the SOMAARTH DDESS, the

regions of focus, are in this domain. The emission scenarios were modelled over three time periods to coincide with the available INCLEN observation data.

We used GEOS-Chem v11-02c, a global chemical transport model driven by assimilated meteorological observations from the NASA Goddard Earth Observing System -- Fast Processing

(GEOS-FP) of the Global Modeling and Assimilation Office (GMAO), to simulate the boundary conditions for the CMAQ modeling. Simulations are performed at 2˚x2.5˚ horizontal resolution with 72 vertical layers, including both the full tropospheric chemistry with complex SOA formation (Marais et al., 2016) and UCX stratospheric chemistry (Eastham et al., 2014). Emissions used the standard HEMCO configuration (Keller et al., 2014), including EDGAR v4.2 anthropogenic

emissions (http://edgar.jrc.ec.europa.eu/overview.php?v=42), biogenic emissions from the MEGAN v2.1 inventory (Guenther et al., 2012), and GFED biomass burning emissions (http://www.globalfiredata.org). Simulations were run for 1 year, after which hourly time series diagnostics were compiled for the CMAQ modeling period. Using the PseudoNetCDF processor, we remapped a subset of the 616 GEOS-Chem-produced species to CMAQ species

(https://github.com/barronh/pseudonetcdf). The resulting ICs and BCs include 119 gas- and particle-phase species, 80 adapted from GEOS-Chem and the remaining 39 (including OH, $HO_2$, ROOH, oligomerized secondary aerosols, coarse aerosol, and aerosol number concentration distributions) from the CMAQ default initial and boundary conditions data (which were developed to represent typical clean-air pollutant concentrations in the United States). Each

CMAQ parent simulation, regardless of study period or emission scenario, used the same GEOS-Chem output dataset for September 2015 (Table 6). The simulations of 9/7/2016 – 9/30/2016 therefore assume each day of September 2016 has the same conditions as those of September 2015. For the CMAQ study period of 12/01/2015 – 12/31/2015, we assume each day has the same boundary conditions as that of 9/30/2015, the final day of the GEOS-Chem simulations.



Each CMAQ domain is simulated separately, such that the nested domain adapts its Ics and BCs from the parent domain simulation results. Thus, the boundary conditions of each domain are time varying and spatially resolved.

5       WRF is a state-of-the-art three-dimensional atmospheric modelling system, developed for numerical weather forecasting and meteorological research. The meteorological simulation was performed for the same domains and time periods as CMAQ (Table 6) and used analyses with incorporated observations prepared by the National Centers for Environmental Prediction (NCEP) for initial and boundary conditions. Parameters were obtained from the National Center for
10 Atmospheric Research (NCAR) on a 0.25° by 0.25° grid for every 6 h and include surface pressure, sea level pressure, geopotential height, temperature, sea surface temperature, soil values, relative humidity, winds, vertical motion, vorticity, and ozone. WRF additionally requires geographic data describing topography and land use. These parameters were obtained from the University Corporation for Atmospheric Research (UCAR). The Thompson scheme was used for
15 the microphysics option, Rapid Radiative Transfer Model G (RRTMG) scheme for longwave and shortwave radiation, Eta similarity for surface layer, and the Noah Land Surface Model: Unified NCEP/NCAR/AFWA scheme with 4 moisture and soil temperature layers. Meteorological outputs from WRF are prepared as inputs to CMAQ by the Meteorology-Chemistry Interface Processor (MCIP).

      Within the chemical transport portion of CMAQ, there are two primary components: a gas-phase chemistry module and an aerosol chemistry, gas-to-particle conversion, module (Figure 4). The present study employs a CMAQ-adapted gas-phase chemical mechanism, CB6R3 (derived from the Carbon Bond Mechanism 06) (Yarwood et al., 2010), and the aerosol-phase
25 mechanism, AERO6, which define the gas-phase and aerosol-phase chemical resolution. The present study considers 70 non-methane hydrocarbon (NMHC) compounds lumped into 12 groups of volatile organic compounds (VOCs). The emissions inventory provides emission rates for 28 chemical species, including 18 gas-phase species and 10 particle-phase species. The CB6R3 adaptation describes atmospheric oxidant chemistry with 127 gas-phase species and 220 gas-





phase reactions, including chlorine and heterogenous reactions. The CMAQ aerosol module (AERO6) describes aerosol chemistry and gas-to-particle conversion with 12 traditional SOA precursor classes, and 10 semi-volatile primary organic aerosol (POA) precursor reactions. The majority of the gas-phase organic species are apportioned to lumped groups by their carbon bond

characteristics, such as single bonds, double bonds, ring structure, and number of carbons. Some organic compounds are apportioned based on reactivity, and others, like isoprene, ethene, and formaldehyde, are treated explicitly.

The secondary organic aerosol module, AERO6, developed specifically for CMAQ,
interfaces with the gas-phase mechanism, predicts microphysical processes of emission, condensation, evaporation, coagulation, new particle formation, and chemistry, and produces a particle size distribution comprising the sum of the Aitken, Accumulation, and Coarse log-normal modes (Figure 5). AERO6 predicts the formation of SOA from anthropogenic and biogenic volatile organic compound (VOC) precursors, as well as semi-volatile POA and cloud processes. CB6R3
accounts for the oxidation of the first-generation products of the anthropogenic lumped VOCs: high-yield aromatics, low-yield aromatics, benzene, PAHs, and long-chain alkanes (Pye and Pouliot, 2012).

In addition to SOA formation from traditional precursors, CMAQv5.2 accounts for the
semi-volatile partitioning and gas-phase aging of POA using the volatility basis set (VBS) framework independently from the rest of AERO6 (Murphy et. Al., 2017). The module distributes directly emitted POA (as the sum of primary organic carbon, POC, and noncarbon organic matter, NCOM) from the emissions inventory input into five new emitted species grouped by volatility: LVPO1, SVPO1, SVPO2, and SVPO3, and IVPO1 (where LV is low volatility, SV is semi-volatile, IV is
intermediate volatility, and PO is primary organic). POA is apportioned to these lumped vapor species using an emission fraction and are oxidized in CB6R3 by OH to LVOO1, LVOO2, SVOO1, SVOO2, and SVOO3 (where OO denotes oxidized organics) with stoichiometric coefficients derived from the 2D-VBS model.  AERO6 then partitions the semi-volatile primary organics and their oxidation products to the aerosol phase. Thus, the treatment of POA as semi-volatile





products leads to an additional twenty species, a particle- and vapor-phase component for each primary organic and oxidation product (Murphy et al., 2017).

Emissions inventory modifications were required to match the most recent aerosol
module, AERO6, in the CMAQ model. Initially, the lumped emissions of PAR (a lumped VOC group characterized by alkanes) and XYL (a lumped VOC group characterized by xylene) derived from grouping specific NMHCs, calculated using the University of Illinois estimation and the Fleming et al. (2018a) emission factors, accounted for characteristics of naphthalene (NAPH) and SOA-producing alkanes (SOAALK), which are not individually described by any of the sources used to
construct the inventory. Moreover, only a subset of VOCs in the smoke could be measured. SVOCs could not be quantified, but they likely make a large contribution to SOA formation. AERO6 predicts the formation of SOA from NAPH and SOAALK independently as well as from XYL and PAR; these secondary aerosol precursor emission rates are calculated with:

$$\mathrm{XYLMN} = 0.998 * \mathrm{XYL}$$
$$\mathrm{NAPH} = 0.002 * \mathrm{XYL}$$
$$\mathrm{PAR_{CMAQ}} = \mathrm{PAR_{calculated}} - 0.00001 * \mathrm{NAPH}$$
$$\mathrm{SOAALK} = 0.108 * \mathrm{PAR_{CMAQ}}$$

where XYLMN, NAPH, PARCMAQ, and SOAALK are the new inventory species (Pye and Pouliot, 2012).  SOA-producing alkanes are treated separately in AERO6. A single daily-averaged rate for each of isoprene (0.8121 moles s$^{-1}$) and terpenes (0.8067 moles s$^{-1}$) was applied as the respective total emissions from all sectors to all computational cells. Isoprene, emitted only by the residential sector, was calculated as the difference of the total rate, predicted by GEOS-Chem,
and the emission rate from all sectors except the residential, already present in the Sharma et al. (2015) inventory. Terpene emissions are assumed to occur only in non-residential source-sectors.



**Ozone**

The 8-hour India Central Pollution Control Board (CPCB) standard for ozone is 100 µg m$^{-3}$ for an 8-hour average. In the alternative unit of ozone mixing ratio, a mass concentration of ozone of 100 µg m$^{-3}$ at a temperature of 298 K at the Earth's surface equates to a mixing ratio of 51 parts-per-billion (ppb).

Several atmospheric modeling studies of ozone levels over India exist (e.g. Kumar et al., 2010).  Kumar et al. (2012) simulated ozone levels using the WRF-Chem model over South Asia for 2008. Chatani et al. (2014) modelled ozone levels over South and East Asia. Sharma et al. (2016) used the same WRF-CMAQ set of models as used by Chatani et al. (2014) with higher resolution emission and meteorological inputs (36 km x 36 km and 25 vertical layers) to assess source and species sensitivities of ozone formation to different precursors over India, some neighboring countries, and the Indian Ocean. The emission sectors considered were residential, transport, industries, power, and non-energy, under the fuel categories of coal, natural gas, diesel, gasoline, liquified petroleum gas, and biomass. The baseline emissions inventory was developed for the year 2010.  Monthly simulations of ozone mixing ratios varied over the range of 30 to 70 ppb; levels were higher in north India during the period of March to June owing to higher solar radiation. Monthly mean ozone levels in rural southern India in 2012 varied between 29 ppb in August during the monsoon season and 56 ppb in April (Reddy et al., 2012). During the July and August monsoon season, ozone levels declined owing to enhanced cloud cover. The Indo-Gangetic plains exhibited the highest ozone concentrations as a result of stronger emission sources.

Sharma et al. (2016) carried out baseline CMAQ simulations for 2010 and compared predictions with measurements at six monitoring locations in India (Thumba, Gadanki, Pune, Anantpur, Mt. Abu, and Nainital). Also carried out were sensitivity simulations in which each emissions sector (transport, domestic, industrial, power, etc.) was systematically set to zero. The domestic sector was found to contribute ~60% of the non-methane volatile organic carbon emissions, followed by 12% from transportation and 20% from solvent use and the oil and gas



sector. The overall $NO_x$-to-VOC mass ratio in the region simulated by Sharma et al. (2016) was 0.55, quite low in comparison to China or the developed world. This exceptionally low $NO_x$-to-VOC ratio was attributed, in part, to the widespread use of biomass fuel for cooking (leading to high VOC emissions), coupled with relatively low $NO_x$ emissions. (Although vehicle emissions are high in urban areas, overall vehicle ownership is relatively low at the national level. In addition, Euro equivalent norms have led to reduction of $NO_x$ emissions.) Predicted $O_3$ levels at the six observation sites tended to exceed measured values, with the ratio of predicted to observed annual average $O_3$ being in the range of 1.04–1.37 at the six locations.

The overall low $NO_x$-to-VOC ratios in India lead to $NO_x$-sensitive $O_3$ formation conditions. Based on emissions inventories, the overall anthropogenic $NMVOC/NO_x$ mass emissions ratio in India in 2010 as computed by Sharma et al. (2016) was 1.82. Considering only ground-level sources, the ratio increases to 3.68. Reduction of overall $NO_x$ emissions computationally by 50% was found to result in a 9% reduction in ambient $O_3$ levels.

Predicted and observed ozone mass concentrations based on CMAQ 4 km resolution simulations at New Delhi over the periods 9/8/2015 – 9/28/2015, 12/3/2015 – 12/31/2015, and 9/8/2016 – 9/30/2016 in the present study are shown in the three panels in Figure 6. Over this period, observed ozone mass concentrations reached levels as high as 300 µg m$^{-3}$. Measured $O_3$ levels in September 2015 in New Delhi (Figure 6) reached a peak of ~200 µg m$^{-3}$ on September 17 and 18, a value that is predicted identically by the model. During the December 2015 period, predicted $O_3$ levels in the early part of the month are underpredicted, significantly during December 7–9, whereas at the end of the month, predicted levels match closely those observed. Finally, in September 2016, $O_3$ levels are overpredicted in the September 8–14 period and generally predicted closely during the latter part of the month. This overall comparison of predictions and observations would appear to be driven by the accuracy of the meteorological fields generated by the model. In general, the degree of agreement between predicted and observed $O_3$ levels in New Delhi over these periods should be considered as reasonable.

**Particulate Matter**

Figures 7 and 8 show measured and predicted total $PM_{2.5}$ at SOMAARTH HQ and compare the results of the computations carried out at 4 km and 1 km resolution, as well as computations for

each emission scenario. Simulations for both time periods capture the general trend well, with two daily peaks and lows aligning with ambient observations. In both cases, however, the highest levels of $PM_{2.5}$ are underestimated, while the atmospheric behavior is more closely predicted for September 2016 than September 2015. $PM_{2.5}$ predictions are similar for both months, with typical values between 15 μg m$^{-3}$ and 100 μg m$^{-3}$. In the upper panels we see that the finer

resolution computations generally predict somewhat higher $PM_{2.5}$ concentrations than the coarser resolution computations; thus a finer resolution contributes to slightly improved agreement with measurements. The closeness of the 4 km and 1 km simulations reflects the closeness of the respective inventories. The lower panels focus on the periods during which observations were available and contrast the "total" and "non-residential" emission scenarios.

The area between the predictions using a total emission inventory and an inventory with zero residential emissions can be considered as the contribution of household sources to ambient $PM_{2.5}$, which is maximized at the peaks. Underpredictions of peak $PM_{2.5}$ concentrations in September could also result because the emission inventory does not account for variations in daily emissions, especially in the agricultural burning sector in which emissions can change

significantly on a daily basis.

Figures 9 and 10 show measured and predicted total $PM_{2.5}$ at Bajada Pahari and the effects of grid resolution and emission scenario. The general trends for December 2015 and September 2016 are again captured. Observations show higher $PM_{2.5}$ levels in December 2015 at

Bajada Pahari than in September 2015 and SOMAARTH HQ in September 2015 and 2016. Typical values reported for December 2015 are between 30 μg m$^{-3}$ and 150 μg m$^{-3}$, owing to frequent temperature inversions in winter and increased emissions from heating. As simulations presented here use a single emissions inventory prepared primarily from data for August and October 2015 (Table 6), emissions from heating in December are underestimated, contributing





to the 30% to 60% discrepancy between predictions and observations at peaks. Bajada Pahari and SOMAARTH HQ are in adjacent 4 km computational grid cells, thus their total $PM_{2.5}$ levels and trends are very similar.

5      Measurements and predictions of $PM_{2.5}$ in New Delhi are shown in Figure 11. Observed and predicted $PM_{2.5}$ levels in New Delhi can exceed 500 μg m$^{-3}$, especially in winter. In this highly populated urban environment, particulate matter levels are more than double those reported in the nearby rural areas. The employed emissions inventory specifies particulate matter surface emissions, which surpass those of Bajada Pahari and SOMAARTH HQ more than 30-fold (Table 3). 10     For September 2015 and 2016, simulation results show poorer agreement (in the form of substantial overestimates) to New Delhi measurements than the rural counterparts, while an improved fit is seen in December 2015 (despite underestimated heating emissions). This suggests the discrepancies between the predictions and observations are at least in part a result of limitations of the emissions inventory. The residential sector inventory was derived 15     independently with a native resolution of 30 arc-seconds (~1 km grid) from the remaining sectors with a native resolution of 36 km. Urban environments have greater diversity of energy-use activities and emissions characteristics than rural environments, where energy-use activities are primarily residential. Therefore, the full emissions inventory in rural areas is represented with generally higher spatial resolution, while the full emissions inventory in cities is comprised with 20     lower spatial resolution. This leads to more accurate predictions in Bajada Pahari and SOMAARTH HQ than in New Delhi.

The significance of household emissions on outdoor $PM_{2.5}$ concentrations is further addressed in Figure 12. Here, the fractional impact of the residential sector on $PM_{2.5}$ is calculated 25     as

$$1 - \frac{PM_{2.5,Non-Residential}}{PM_{2.5,Total}}$$

where $PM_{2.5,Non-Residential}$ and $PM_{2.5,Total}$ are the predictions from the Non-Residential and Total emission scenario simulation, respectively (Table 5). The contribution of household emissions to



total PM$_{2.5}$ is strongly correlated with meal times and maximum contributions in Bajada Pahari and SOMAARTH HQ are more than double that in New Delhi. In September 2015 and 2016, household energy-use activities account for up to 33% of ambient PM$_{2.5}$ at SOMAARTH HQ and up to 28% at Bajada Pahari in September 2016. As the residential emissions inventory was
prepared for the month of August, there are no emissions from heating, and therefore simulations predict lower significance of the residential sector in December 2015 than expected.

Figures 13–16 show predictions of secondary organic PM$_{2.5}$ in each CMAQ simulation, the contribution of secondary organic matter (SOM) to total PM$_{2.5}$, and the influence of the
residential sector. Figures 13 and 14 compare secondary organic PM$_{2.5}$ predicted in the "total" and "non-residential" emissions scenarios carried out at 4 km resolution at SOMAARTH HQ, Bajada Pahari, and New Delhi Punjabi Bagh. Like PM$_{2.5}$, SOM is typically predicted to be higher in New Delhi than in the rural sites, as well as in December over September, due to higher PM$_{2.5}$ and SOA precursor emissions and ambient concentrations in urban environments and longer
residence times allowing for more aging during winter. SOM predictions reach typical values of 2.2 µg m$^{-3}$ to 8.0 µg m$^{-3}$ in New Delhi and 2.0 µg m$^{-3}$ to 7.0 µg m$^{-3}$ at the rural sites. The contribution of SOM to total PM$_{2.5}$ (Figure 15), however, is predicted to be lower in New Delhi during December, while SOM accounts for up to 19% of PM$_{2.5}$ at the rural sites during September months. As the emissions inventory used for each simulation is the same, the difference in the
significance of SOM is likely due to substantially higher primary PM$_{2.5}$ concentrations in December than September, while SOM remains comparable in both months. Similarly, the fractional impact of the residential sector on SOM (Figure 16) is lowest in New Delhi, accounting for up to 12% in New Delhi during September months when accounting for up to 23% in SOMAARTH HQ and Bajada Pahari. While the meal time signature is still present in New Delhi,
this difference is attributable to the greater significance of non-residential SOA precursor emissions in the city.

The CMAQ model predicts the composition of secondary organic aerosol (see Figure 5). The predictions of CMAQ can be compared with the estimates of Fleming et al. (2018b) based on





direct emission measurements carried out at a village kitchen in Khatela, Palwal District between August 5 and September 3, 2015. Fleming et al. (2018b) quantified emissions of CO, $CO_2$, and 76 VOCs from cook fires in a village home, employing two traditional cookstoves, the *chulha* and the *angithi*, using two kinds of biomass fuel, brushwood and dung cakes. Emission factors were

calculated using the carbon-balance method, which assumes that all carbon in the fuel is converted to $CO_2$, CO, VOCs, and particulate matter (PM) when the fuel is burned (Smith et al., 2000). Alkanes, alkenes, aromatics, oxygenates, halogen- and sulfur-containing compounds all exhibited higher emissions per kg of fuel carbon when dung fuels and *angithi* stoves were utilized compared to brushwood fuels and *chulha* stoves, respectively. Fleming et al. (2018b) estimated

SOA production from the different cook fire types based on the measured VOCs emitted, using secondary organic aerosol potential (SOAP) values from Derwent et al. (2010), computed via an air parcel model travelling across Europe. Gas-phase chemistry in the parcel was computed by the Master Chemical Mechanism (http://mcm.leeds.ac.uk/MCM). SOA mass was calculated based on equilibrium partitioning of oxidation products to aerosol. Fleming et al. (2018b)

determined that fuel type is more important than stove type in terms of SOA formation. Benzene emissions were estimated to be responsible for at least one-half of the SOA formation from the VOCs emitted by the *chulha* cook fire.  On the whole, aromatics were estimated to comprise ~95% of SOA precursors for all cook fires.

**Conclusions**

Air quality in India is determined by a mixture of industrial and motor vehicle emissions, and anthropogenic fuel combustion, that includes residential burning of biomass for household uses, such as cooking. Average daily $PM_{2.5}$ levels frequently exceed the 24-hour standard of 60 μg m$^{-3}$

and can exceed 200 μg m$^{-3}$, even in rural areas. $PM_{2.5}$ is a mixture of directly-emitted particulate matter and that formed by the atmospheric conversion of volatile organic compounds to secondary organic aerosol.



In India, over 50% of households report use of wood, crop residues, or dung as cooking fuel; such fuels produce significant gas- and particle-phase emissions. Here, we assess the extent to which observed $O_3$ and $PM_{2.5}$ levels in India can be predicted using state-of-the-science emissions inventories and atmospheric chemical transport models. We have focused on the 308

sq km of the SOMAARTH Demographic, Development, and Environmental Surveillance Site (DDESS) in the Palwal District of Haryana, India.

Simulations of ozone levels in New Delhi reported here are largely in agreement with ambient monitoring data, although the simulations fail to capture several one- to two-day ozone

episodes that exceed predictions by a factor of two or more. The overall good agreement between observed and predicted $O_3$ levels, also demonstrated in the study of Sharma et al. (2016), suggests that gas-phase atmospheric chemistry over India is reasonably well understood.

Atmospheric simulation of particulate matter levels over a complex region like India tends

to be demanding, owing to the combination of a wide range of primary particulate emissions and the presence of secondary organic matter from atmospheric gas-phase reactions generating low-volatility gas-phase products that condense into the particulate phase, forming *secondary organic aerosol* (SOA).  Consequently, the main focus of the present work has been the evaluation of the extent to which ambient particulate matter levels over the current region of India can be

predicted. Simulations capture the general trend of observed daily peaks and lows of particulate matter, with $PM_{2.5}$ reaching values as high as 250 µg m$^{-3}$. The fractional contribution of secondary particulate matter to total $PM_{2.5}$ mass is found to average somewhat less than 10%, indicating that primary particulate matter emissions dominate in the region of study. Finally, we evaluated the fractional impact of the residential sector emissions on the formation of secondary organic

aerosol, as a function of time of day, for New Delhi, SOMAARTH HQ, and Bajada Pahari. Moreover, the fractional household contribution in New Delhi barely exhibited a diurnal pattern. On the other hand, the predicted fractional contribution of residential sector emissions to secondary organic $PM_{2.5}$ in Bajada Pahari and SOMAARTH HQ reaches values as high as 23% and,

moreover, displays a distinct diurnal profile, with maxima corresponding to the morning and evening mealtimes.

Air quality studies such as the present one provide a quantification of the elements of atmospheric composition in India, especially that owing to household sources. The importance of replacing traditional household combustion devices with modern technology is strongly evident in studies such as the present one.

## Author Contributions

BR developed the model code, carried out the simulations, and wrote the paper; RZ and KB assisted with the simulations; AP designed the experiments; SS, SK, TB, NL, BO and LX helped formulate the emissions inventory; LF, RN, SM, and DB designed and carried out measurements; SN, RE, AY, and NA performed data analysis; KS designed the research; JS designed the research

and wrote the paper.

## Acknowledgment

This work was supported by EPA STAR grant R835425 Impacts of Household Sources on Outdoor
Pollution at Village and Regional Scales in India. The contents are solely the responsibility of the authors and do not necessarily represent the official views of the US EPA.



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




**Table 1.** Residential Emissions Inventory Sources by Species

| | CMAQ Required Species[1] | Source | | Solely Emitted by Residential Sector |
|---|---|---|---|---|
| **Gas** | NO | T. Bond (University of Illinois) $NO_x$ using Sharma et al. (2015) $NO:NO_2 = 10:1$ | | No |
| | $NO_2$ | | | No |
| | $SO_2$ | Sharma et al. (2015) | | No |
| | $NH_3$ | Sharma et al. (2015), assumed to be negligible | | |
| | CO | T. Bond (University of Illinois) | | No |
| **NMHC** | $ALD_2$ | Speciation from T. Bond (University of Illinois) $PM_{2.5}$ using Fleming et al. (2018a,b) emission factors | | No |
| | $ALD_X$ | | | Yes |
| | ETH | | | No |
| | ETHA | | | No |
| | ETOH | | | No |
| | FORM | | | No |
| | MEOH | | | No |
| | OLE | | | No |
| | $PAR_{calculated}$[3] | | | No |
| | TOL | | | No |
| | $XYL$[3] | | | No |
| **CMAQ AERO6 Species** | **ISOP**[2] | All-sector total ISOP emission from GEOS-Chem daily average and subtracted non-residential ISOP emission from Sharma et al. (2015) | | No |
| | **TERP**[2] | Assumed to be negligible | | |
| | **XYLMN** | XYLMN = 0.998 * XYL | Pye and Pouliot (2012) | No |
| | **NAPH** | NAPH = 0.002 * XYL | | No |
| | $PAR_{CMAQ}$ | $PAR_{CMAQ}$ = $PAR_{calculated}$ - 0.00001 * NAPH | | No |
| | **SOAALK** | SOAALK = 0.108*$PAR_{CMAQ}$ | | No |
| **PM** | $P_{EC}$ | T. Bond (University of Illinois) | | No |
| | $P_{OC}$ | | | No |
| | $P_{NA}$ | Speciation from T. Bond (University of Illinois) $PM_{2.5}$ using Jayarathne et al. (2018) | | Yes |
| | $P_{CL}$ | | | Yes |
| | $P_K$ | | | Yes |
| | $P_{NH4}$ | | | Yes |
| | $P_{NO3}$ | | | No |
| | $P_{SO4}$ | | | No |
| | $PM_{OTHR}$ | $PM_{OTHR}$ = $PM_{2.5}$ - ($P_{NA}$+$P_{NH4}$+$P_K$+$P_{CL}$+$P_{NO3}$+$P_{SO4}$) | | No |
| | $PM_C$ | Sharma et al. (2015) | | No |
| | $P_{NCOM}$ | Unknown, assumed to be 0 | | |
| | $P_{H2O}$ | | | |
| | $P_{AL}$ | Assumed to be negligible | | |
| | $P_{CA}$ | | | |
| | $P_{FE}$ | | | |
| | $P_{MG}$ | | | |
| | $P_{MN}$ | | | |
| | $P_{SI}$ | | | |
| | $P_{TI}$ | | | |

[1]Bolded species contribute to SOA production via the AERO6 module. [2]Total isoprene and terpene emissions from all sectors are taken from GEOS-Chem and were included only in the $O_3$ simulations. [3]$PAR_{calculated}$ and XYL are excluded from CMAQ and replaced with $PAR_{CMAQ}$, XYLMN, NAPH, and SOAALK.





**Table 2.** Particulate Matter Surface Emissions over Parent Domain

| Species | | Emission Rate | % Emitted by Residential Sector |
|---|---|---|---|
| **Particulate Matter (kg/day)** | $P_{OC}$ | $1.48 \times 10^6$ | 30.78 |
| | $P_{EC}$ | $7.18 \times 10^5$ | 15.89 |
| | $P_{CL}$ | $1.69 \times 10^3$ | 100 |
| | $P_K$ | $4.61 \times 10^3$ | 100 |
| | $P_{NA}$ | $2.46 \times 10^4$ | 10.07 |
| | $P_{NH4}$ | $2.11 \times 10^5$ | 1.47 |
| | $P_{NO3}$ | $6.51 \times 10^5$ | 27.90 |
| | $P_{SO4}$ | $1.18 \times 10^6$ | 61.45 |
| | $PM_C$ | $9.00 \times 10^3$ | 100 |
| | $PM_{OTHR}$ | $2.18 \times 10^4$ | 100 |
| **SOA Precursor VOCs (mol/day)** | $NAPH$ | $6.82 \times 10^3$ | 2.72 |
| | $SOAALK$ | $3.75 \times 10^6$ | 34.54 |
| | $TOL$ | $1.54 \times 10^6$ | 27.21 |
| | $XYLMN$ | $3.40 \times 10^6$ | 2.72 |

**Table 3.** Mealtime[1] Particulate Matter Surface Emissions over Corresponding 16 km[2] Grid Cell

| Species | | Bajada Pahari | | SOMAARTH HQ | | New Delhi | |
|---|---|---|---|---|---|---|---|
| | | Total | % Residential | Total | % Residential | Total | % Residential |
| **Particulate Matter (kg/day)** | $P_{OC}$ | 35.17 | 67.13 | 36.04 | 100 | 609.73 | 5.70 |
| | $P_{EC}$ | 10.22 | 33.23 | 6.02 | 100 | 346.84 | 2.21 |
| | $P_{CL}$ | 2.19 | 100 | 3.40 | 100 | 3.40 | 100 |
| | $P_K$ | 0.43 | 100 | 0.66 | 100 | 0.66 | 100 |
| | $P_{NA}$ | 0.08 | 100 | 0.12 | 100 | 0.12 | 100 |
| | $P_{NH4}$ | 1.037 | 100 | 1.61 | 100 | 1.61 | 100 |
| | $P_{NO3}$ | 0.37 | 32.01 | 0.18 | 100 | 12.59 | 1.45 |
| | $P_{SO4}$ | 2.49 | 5.90 | 0.23 | 100 | 116.80 | 0.20 |
| | $PM_C$ | 63.99 | 91.94 | 72.56 | 100 | 275.99 | 7.12 |
| | $PM_{OTHR}$ | 13.92 | 61.94 | 13.37 | 100 | 276.91 | 4.83 |
| **SOA Precursor VOCs (mol/day)** | $NAPH$ | 0.11 | 6.50 | 0.03 | 59.56 | 3.78 | 0.65 |
| | $SOAALK$ | 112.31 | 50.62 | 113.20 | 77.95 | 1696.26 | 11.14 |
| | $TOL$ | 43.88 | 42.28 | 39.38 | 71.05 | 750.00 | 1.04 |
| | $XYLMN$ | 56.47 | 6.50 | 13.03 | 59.56 | 1886.15 | 0.65 |

5  [1]Mealtimes are assumed to be 5am – 1pm and 5pm – 8pm (local).



**Table 4.** Ambient Observation Data Availability

| Location (Grid Cell) | Timeframe | Measured Species |
|---|---|---|
| Bajada Pahari[1] (74,74) | 12/15/15 – 12/31/15 | PM$_{2.5}$ |
| | 9/19/16 – 9/30/16 | |
| SOMAARTH HQ[1] (75,74) | 9/22/15 – 9/26/15 | |
| | 9/23/16 – 9/30/16 | |
| New Delhi[2] (70,92) | 9/7/15 – 9/27/15 | PM$_{2.5}$, O$_3$, NO$_2$ |
| | 12/1/15 – 12/31/15 | |
| | 9/7/16 – 9/30/16 | |

[1]Data from the International Epidemiological Clinical Network. Observations at Bajada Pahari are the average of two monitoring

5    locations that coincide within the same grid cell. [2]Data from the Central Pollution Control Board of India at New Delhi Punjabi Bagh monitoring station.





**Table 5.** Emission Simulation Scenarios

| Emission Scenario | Description | Domain (Resolution) | Boundary and Initial Conditions |
|---|---|---|---|
| Total | Includes emissions from all source-sectors, represents the overall atmospheric environment | Parent (4km) | GEOS-Chem "total"[1] |
| | | Nested (1km) | Total scenario parent domain output |
| Non-Residential | Includes emissions from all source-sectors except the residential, represents "turning off" household emissions | Parent (4km) | GEOS-Chem "total"[1] |
| | | Nested (1km) | Non-Residential scenario parent domain output |

[1]GEOS-Chem v11-02c was run with a "total" emission inventory to produce the boundary and initial conditions for the parent domain simulations of both the Total and Non-Residential emission scenarios.

**Table 6.** CMAQ Simulations

| CMAQ Simulations | | 09/07/15 – 09/27/15 | 12/01/15 – 12/31/15 | 09/07/16 – 09/30/16 |
|---|---|---|---|---|
| Meteorology Simulations (WRF) | | 09/07/15 – 09/27/15 | 12/01/15 – 12/31/15 | 09/07/16 – 09/30/16 |
| Boundary Conditions[1] (GEOS-Chem) | | 09/07/15 – 09/27/15 | 09/30/15 for all days | 09/07/15 – 09/30/15 |
| Initial Conditions (GEOS-Chem) | | 09/07/15 | 09/30/15 | 09/07/15 |
| Emission Inventory | $PM_{2.5}$ Simulations | Prepared from primarily August and October 2015 data, without isoprene and terpenes | | |
| | $O_3$, $NO_2$ Simulations | Prepared from primarily August and October 2015 data, with isoprene and terpenes | | |

[1]Boundary conditions for parent domain only are listed here. The nested domain boundary conditions are taken from the concentration outputs from the parent domain CMAQ simulation.



**Table 7.** Properties of anthropogenic traditional semi-volatile SOA precursors in CMAQv5.2

| SOA species | Precursor | Oxidants | Semi-volatile | α (mass-based) | C* (μg/m³) | ΔH$_{vap}$ (kJ/mol) | # of C | Molecular weight (g/mol) | OM/OC |
|---|---|---|---|---|---|---|---|---|---|
| AALK1 | long-chain alkanes | OH | SV_ALK1 | 0.0334 | 0.15 | 53.0 | 12 | 168 | 1.17 |
| AALK2 | long-chain alkanes | OH | SV_ALK2 | 0.2164 | 51.9 | 53.0 | 12 | 168 | 1.17 |
| AXYL1 | XYLMN | OH,NO | SV_XYL1 | 0.0310 | 1.3 | 32.0 | 8 | 192 | 2.0 |
| AXYL2 | XYLMN | OH,NO | SV_XYL2 | 0.0900 | 34.5 | 32.0 | 8 | 192 | 2.0 |
| AXYL3 | XYLMN | OH,HO₂ | nonvolatile | 0.36 | NA | NA | NA | 192 | 2.0 |
| ATOL1 | TOL | OH,NO | SV_TOL1 | 0.0310 | 2.3 | 18.0 | 7 | 168 | 2.0 |
| ATOL2 | TOL | OH,NO | SV_TOL2 | 0.0900 | 21.3 | 18.0 | 7 | 168 | 2.0 |
| ATOL3 | TOL | OH,HO₂ | nonvolatile | 0.30 | NA | NA | NA | 168 | 2.0 |
| ABNZ1 | benzene | OH,NO | SV_BNZ1 | 0.0720 | 0.30 | 18 | 6 | 144 | 2.0 |
| ABNZ2 | benzene | OH,NO | SV_BNZ2 | 0.8880 | 111 | 18 | 6 | 144 | 2.0 |
| ABNZ3 | benzene | OH,HO₂ | nonvolatile | 0.37 | NA | NA | NA | 144 | 2.0 |
| APAH1 | naphthalene | OH,NO | SV_PAH1 | 0.2100 | 1.66 | 18 | 10 | 243 | 2.03 |
| APAH2 | naphthalene | OH,NO | SV_PAH2 | 1.0700 | 265 | 18 | 10 | 243 | 2.03 |
| APAH3 | naphthalene | OH,HO₂ | nonvolatile | 0.73 | NA | NA | NA | 243 | 2.03 |

The semi-volatile reaction products of "long alkanes" (SV_ALK1 and SV_ALK2) are parameterized by Presto et al. (2010). Values for "low-yield aromatics" products (SV_XYL1 and SV_XYL2) are based on xylene, with the enthalpy of vaporization (ΔH$_{vap}$) from studies of m-xylene and 1,3,5-trimethylbenzene. ΔH$_{vap}$ for products of "high-yield aromatics" (SV_TOL1 and SV_TOL2) are based on the
5  higher end of the range for toluene. The products of benzene (SV_BNZ1 and SV_BNZ2) assume the same value for ΔH$_{vap}$. All semi-volatile aromatic products are assigned stoichiometric yield (α) and effective saturation concentration (C*) values from laboratory measurements by Ng et al. (2007). Remaining parameters for PAH reaction products (SV_PAH1 and SV_PAH2) are taken from Chan et al. (2009). Properties of semi-volatile primary organic aerosol precursors are given in Murphy et al. (2017).


**Figures**

Fig. 1. Geographic area of simulation. The upper left panel shows the entirety of India, and the lower left and right panels show closeups of the model domains. The parent domain (red) spans a 600 km by 600 km area with a grid resolution of 4 km (150 cells along each axis) and includes
both New Delhi and SOMAARTH DDESS. The nested domain (blue) spans a 100 km by 100 km area with a grid resolution of 1 km (100 cells along each axis) and includes SOMAARTH DDESS. New Delhi lies 15 km outside the domain.

Fig. 2. Fraction of daily household emissions by fuel-use activity. Red, green, blue, and purple
indicates cooking, space heating, water heating, and lighting, respectively. This represents the fraction of activity-specific daily emissions at each hour. Each species obeys the same profile. Data source:  University of Illinois.

Fig. 3. Fuel type assumed for household emissions. Left panel: 600 by 600 km parent domain at
4 km resolution. Right panel: 100 km by 100 km nested domain at 1 km resolution. Red indicates cells where dung use dominated emissions and thus was assumed to be the sole fuel type used. Orange indicates cells where wood and agricultural residue use dominated emissions and was thus assumed to be the sole fuel type used. Data source:  University of Illinois.

Fig. 4. Model flow diagram. CMAQ (blue) requires inputs of emissions (green), meteorology (orange), and initial and boundary conditions (purple) to produced gridded airborne concentrations (yellow). [1]GEOS-Chem v11-02c provides initial and boundary conditions for the parent domain, and gridded airborne concentrations are fed back into CMAQ simulations of the nested domain (purple outlines). [2]The emissions inventory is derived for sources shown in Table
25 1.

Fig. 5. Treatment of anthropogenic SOA in CMAQv5.2. Predicted aerosol species are included in the black box. Species in white boxes are semi-volatile and species in gray boxes are nonvolatile. Blue indicates species and processes predicted by CB6R3. All other coloring
indicates the AERO6 mechanism where green arrows are 2-product volatility distribution, orange arrows are particle- and vapor-phase partitioning, and purple arrows are oligomerization. In AERO6, anthropogenic and biogenic VOC emissions (lumped by category), are oxidized by OH, NO, and $HO_2$ and OH, $O_3$, NO, and $NO_3$ respectively, to semi-volatile products that undergo partitioning to the particle phase (Pye et al., 2015).  Semi-volatile
primary organic pathways in CMAQv5.2 are described by Murphy et al. (2017). Potential SOA from combustion (PCSOA) is omitted from simulations presented here.

Fig. 6. Measured (shaded) and predicted (blue) ozone in New Delhi: 9/8/15-9/27/15 (top panel), 12/1/15-12/31/15 (middle panel), and 9/8/16-9/30/16 (bottom panel). CMAQ predictions
shown are from computations carried out at 4 km resolution. The tick marks correspond to midnight, local time.





Fig. 7. Measured (shaded) and predicted (green) PM$_{2.5}$ at SOMAARTH HQ: 9/8/15-9/27/15. In the upper panel, green and black lines correspond to 4 km and 1 km computational resolution, respectively. The lower panel focuses on the period 9/22/15 – 9/27/15 during which observations were available. Here the green line corresponds to the full emission inventory

5          simulation, whereas the black line corresponds to the predictions with zero residential emissions (the "total" and "non-residential" emission scenarios, respectively, as described in Table 5). Both computations were carried out at 4 km resolution. The shaded area represents the ambient observations. Pink bars denote observation peaks that occur during mealtimes (5am–1pm and 5pm–8pm).

Fig. 8. Measured (shaded) and predicted (green) PM$_{2.5}$ at SOMAARTH HQ: 9/8/16-9/30/16. In the upper panel, the green and black lines correspond to 4 km and 1 km computational resolution, respectively. The lower panel focuses on the period 9/24/16 – 9/30/16 during which observations were available. Here the green line corresponds to the full emission inventory

15        simulation, whereas the black line corresponds to the predictions with zero residential emissions (the "total" and "non-residential" emission scenarios, respectively, as described in Table 5). Both computations were carried out at 4 km resolution. The shaded area represents the ambient observations. Pink bars denote observation peaks that occur during mealtimes (5am–1pm and 5pm–8pm).

Fig. 9. Measured (shaded) and predicted (red) PM$_{2.5}$ at Bajada Pahari 12/1/15 -12/31/15. In the upper panel, the red and black lines correspond to 4 km and 1 km computational resolution, respectively. The lower panel focuses on the period 12/16/15-12/31/15 during which observations were available. Here the red line corresponds to the full emission inventory

25        simulation, whereas the black line corresponds to the predictions with zero residential emissions (the "total" and "non-residential" emission scenarios, respectively, as described in Table 5). Both computations were carried out at 4 km resolution. The shaded area represents the ambient observations. Pink bars denote observation peaks that occur during mealtimes (5am–1pm and 5pm–8pm).

Fig. 10. Measured (shaded) and predicted (red) PM$_{2.5}$ at Bajada Pahari 9/8/16-9/30/16. In the upper panel, the red and black lines correspond to 4 km and 1 km computational resolution, respectively. The lower panel focuses on the period 9/20/16-9/30/16 during which observations were available. Here the red line corresponds to the full emission inventory

35        simulation, whereas the black line corresponds to the predictions with zero residential emissions (the "total" and "non-residential" emission scenarios, respectively, as described in Table 5). Both computations were carried out at 4 km resolution. The shaded area represents the ambient observations. Pink bars denote observation peaks that occur during mealtimes (5am–1pm and 5pm–8pm).

Fig. 11. PM$_{2.5}$ at New Delhi 9/7/15 – 9/27/15 (top panel), 12/1/15 – 12/31/15 (middle panel), and 9/8/16 – 9/30/16 (bottom panel) where shading represents ambient observations and blue and black lines correspond to predictions using a full emissions inventory and predictions using





an emissions inventory with zero residential emissions, respectively (Table 5). Computations were carried out at 4 km resolution.

Fig. 12. Average diurnal fractional impact of residential sector emissions on $PM_{2.5}$. Top panel:
SOMAARTH HQ (green)  and New Delhi (blue), September 2015. Middle panel: Bajada Pahari and New Delhi (blue), December 2015. Bottom panel: Bajada Pahari (red), SOMAARTH HQ (green), and New Delhi (blue), September 2016. Shading indicates meal times. Fractional impact of the residential sector is calculated as  $1 - \frac{PM_{2.5,Non-Residential}}{PM_{2.5,Total}}$ , where $PM_{2.5,Non-Residential}$ and $PM_{2.5,Total}$ are the predictions from the Non-Residential and Total emission scenario simulation,
respectively (Table 5), averaged over simulation durations (Table 6). Computations were carried out at 4 km resolution.

Fig. 13. Predicted secondary organic $PM_{2.5}$ at SOMAARTH HQ 9/8/15-9/26/15 (first panel), SOMAARTH HQ 9/8/16-9/30/16 (second panel), Bajada Pahari 12/3/15-12/30/15 (third panel),
and Bajada Pahari 9/8/16-9/30/16 (fourth panel). Red and green lines correspond to predictions using a full emissions inventory, and black lines correspond to predictions using an emissions inventory with zero residential emissions (Table 5). Computations were carried out at 4 km resolution.

Fig. 14. Predicted secondary organic $PM_{2.5}$ at New Delhi 9/8/15-9/26/15 (top panel), 12/3/15-12/30/15 (middle panel), 9/10/16-9/28/16 (bottom panel). Blue and black lines correspond to predictions using a full emissions inventory and predictions using an emissions inventory with zero residential emissions, respectively (Table 5). Computations were carried out at 4 km resolution.

Fig. 15. Contribution of secondary organic matter to $PM_{2.5}$. Top panel: SOMAARTH HQ (green) and New Delhi (blue), September 2015. Middle panel: Bajada Pahari (red) and New Delhi (blue), December 2015. Bottom panel: Bajada Pahari (red), SOMAARTH HQ (green), and New Delhi (blue), September 2016. Contribution of SOM to total $PM_{2.5}$ is defined as the ratio of
predictions of SOM to predictions of total $PM_{2.5}$, using a full emissions inventory. Computations were carried out at 4 km resolution.

Fig. 16. Fractional impact of residential sector on secondary organic matter. Average diurnal fractional impact of residential sector emissions on SOM. Top panel: SOMAARTH HQ (green)
and New Delhi (blue), September 2015. Middle panel: Bajada Pahari (red) and New Delhi (blue), December 2015. Bottom panel: Bajada Pahari (red), SOMAARTH HQ (green), and New Delhi (blue), September 2016. Shading indicates meal times. Fractional impact of the residential sector is calculated as $1 - \frac{SOM_{Non-Residential}}{SOM_{Total}}$ , where $SOM_{Non-Residential}$ and $SOM_{Total}$ are the predictions from the Non-Residential and Total emission scenario simulation, respectively
(Table 5), averaged over simulation durations (Table 6). Computations were carried out at 4 km resolution.




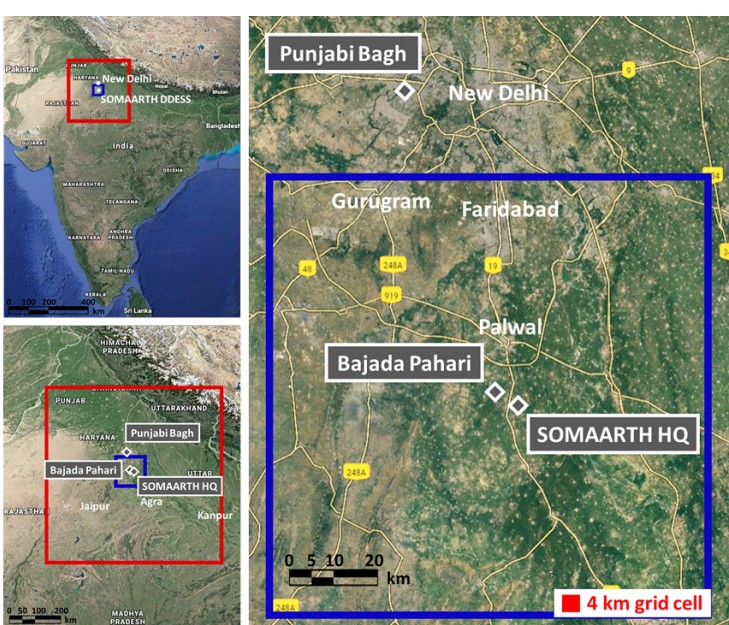

Figure 1.



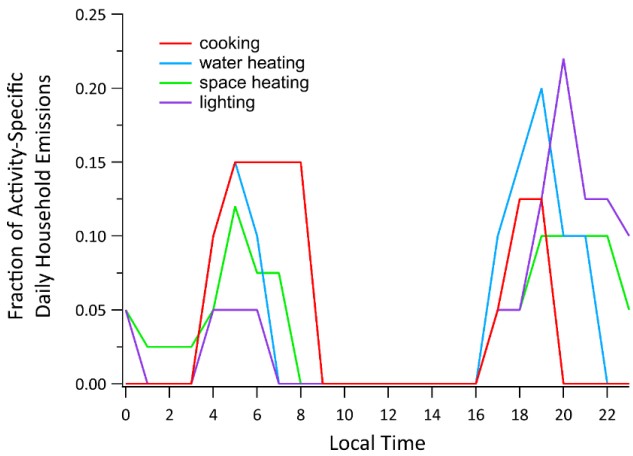

Figure 2.





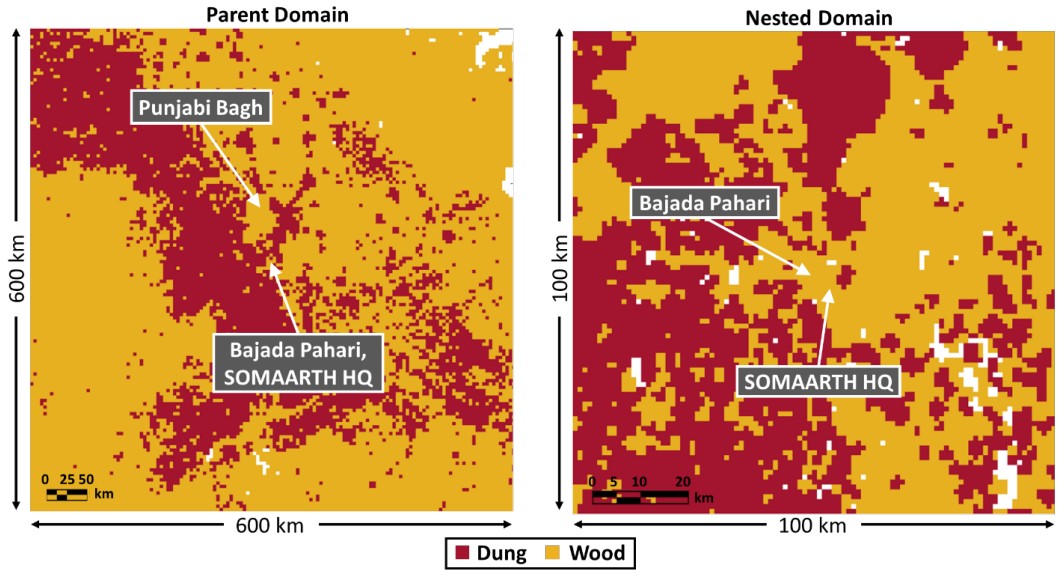

Figure 3.



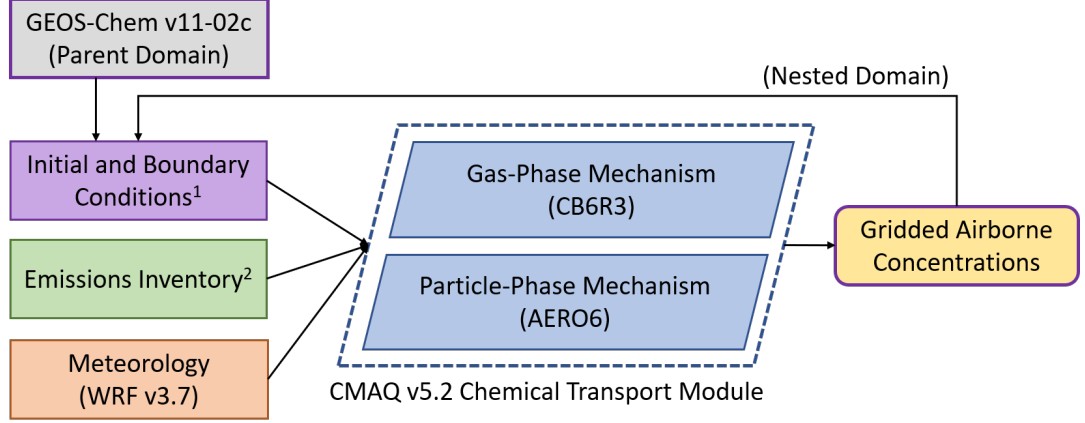

Figure 4.





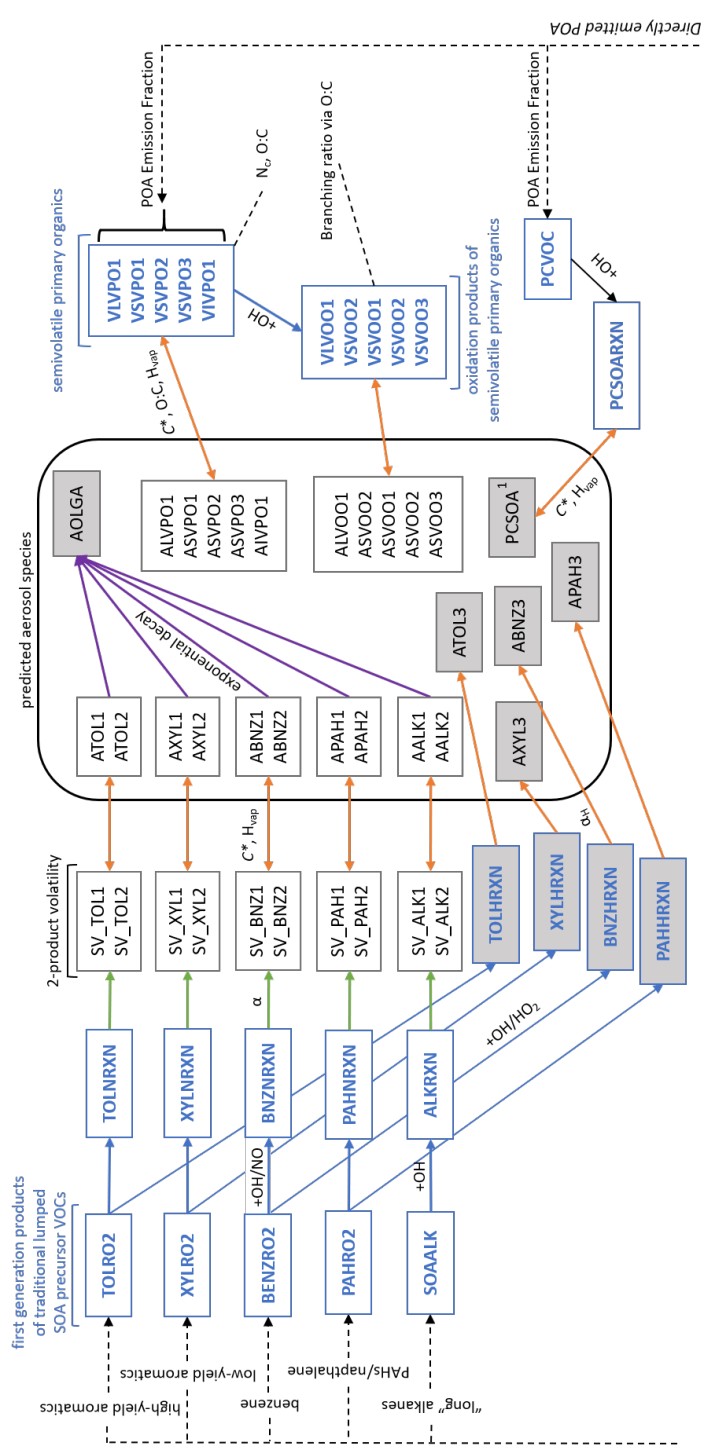

Figure 5.





5    Figure 6.





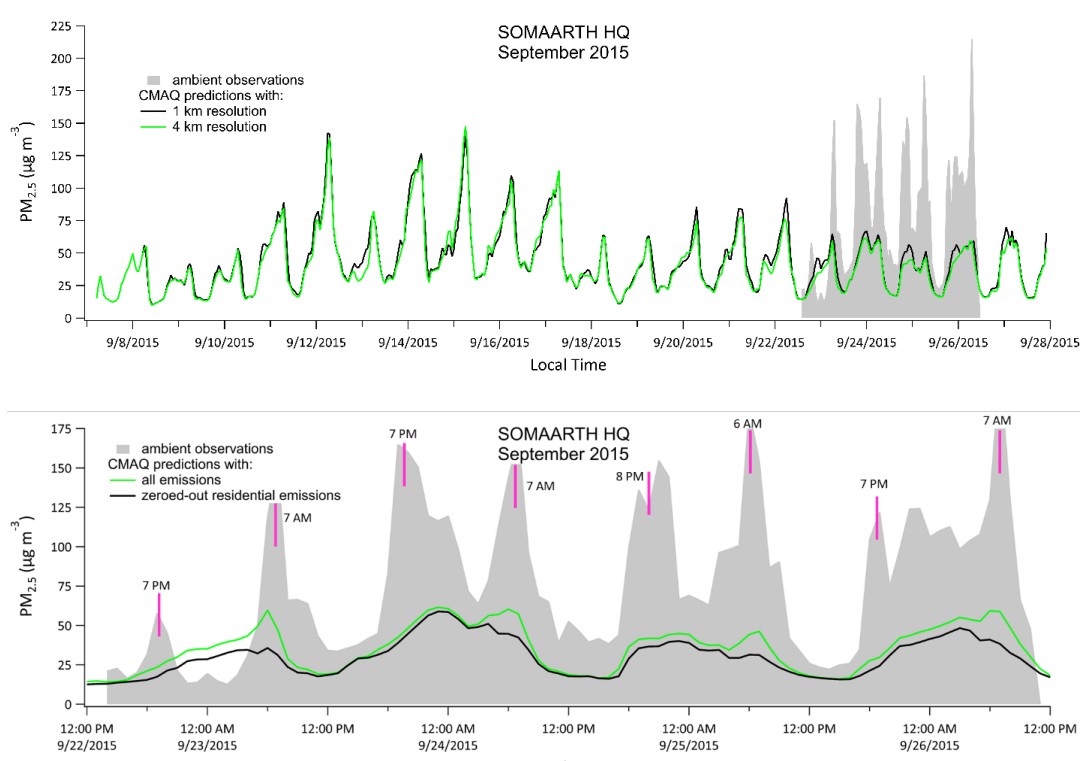

5      Figure 7.



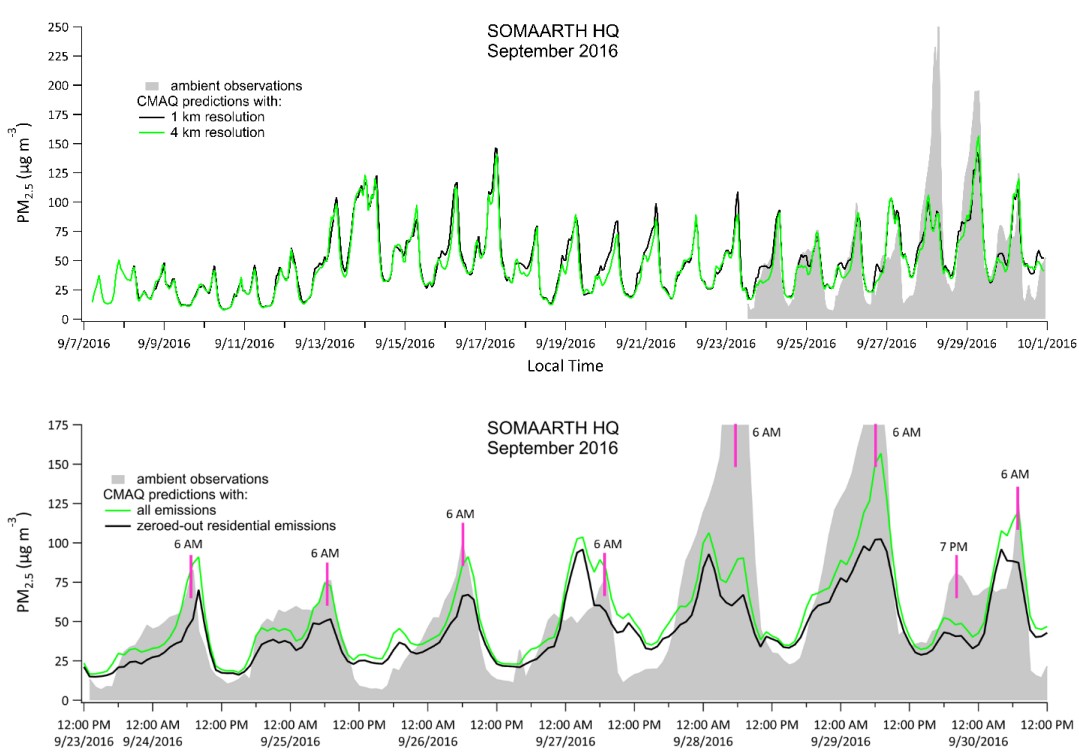

Figure 8.





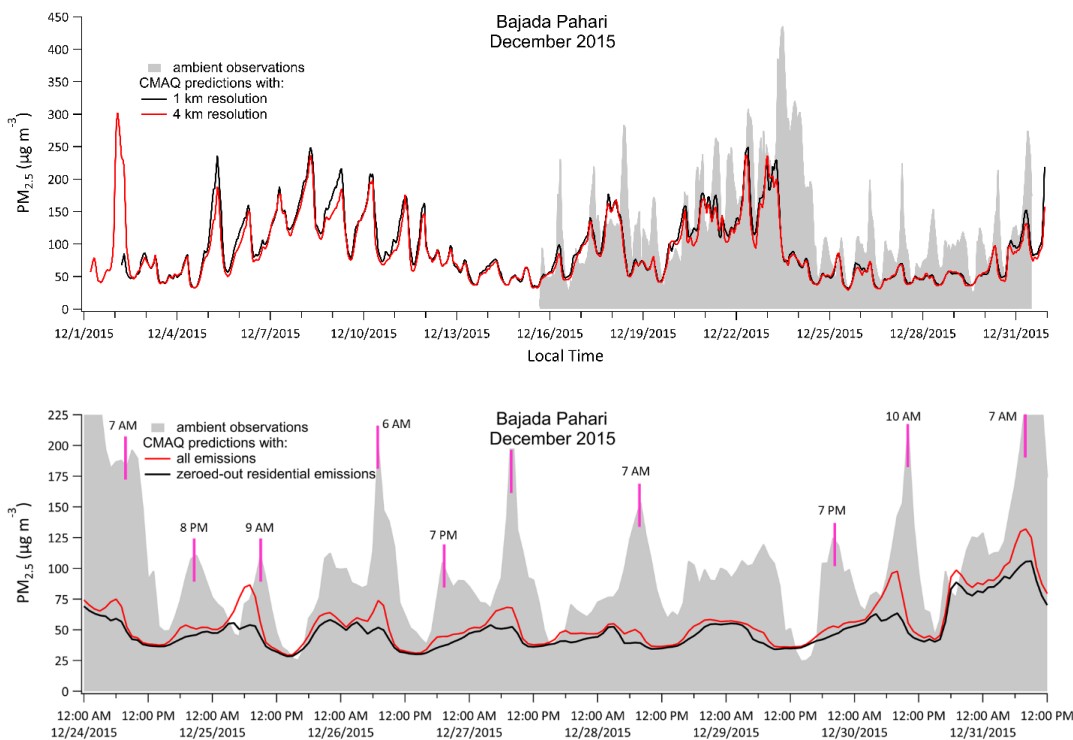

Figure 9.



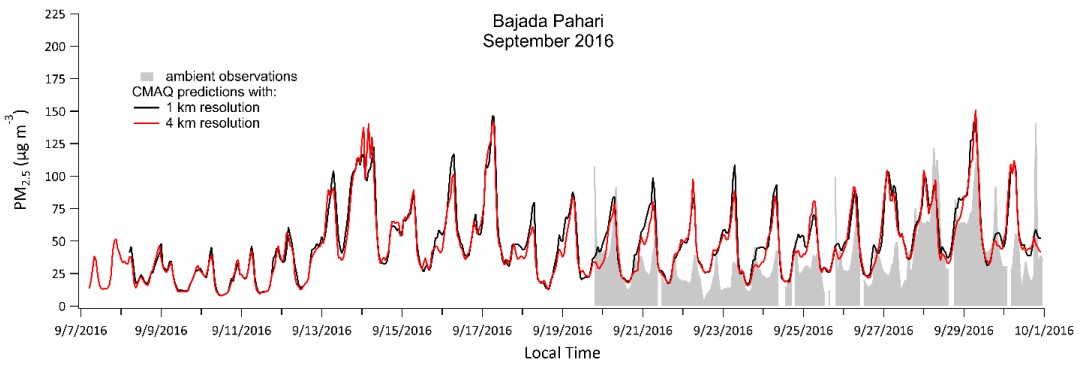

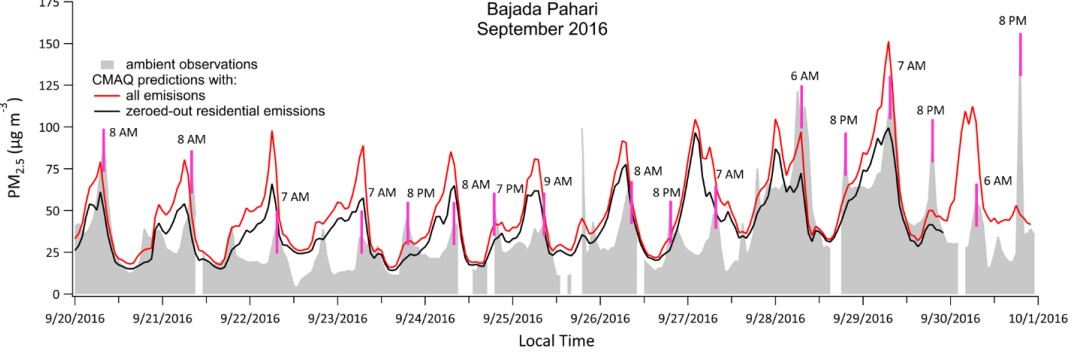

Figure 10.





Figure 11.

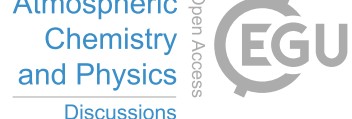

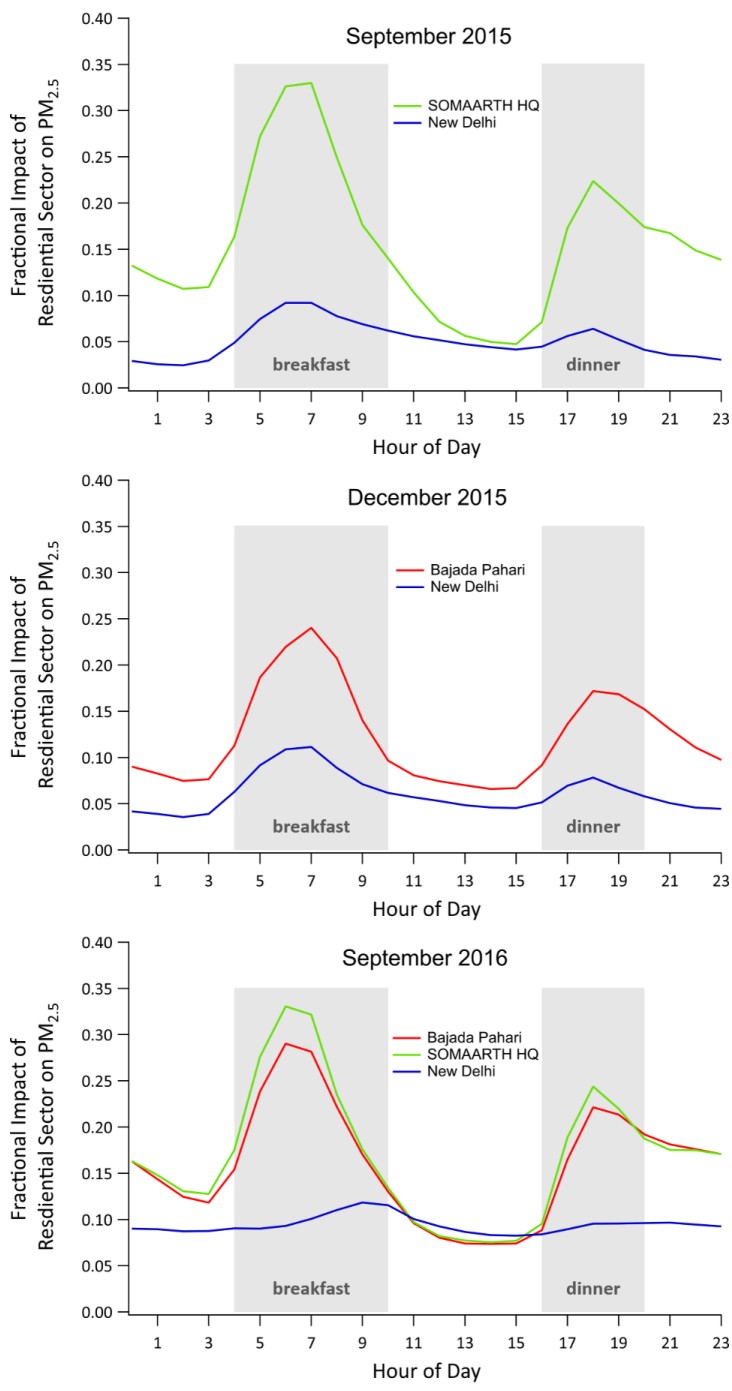

Figure 12.





5      Figure 13.




Figure 14.







Figure 15.



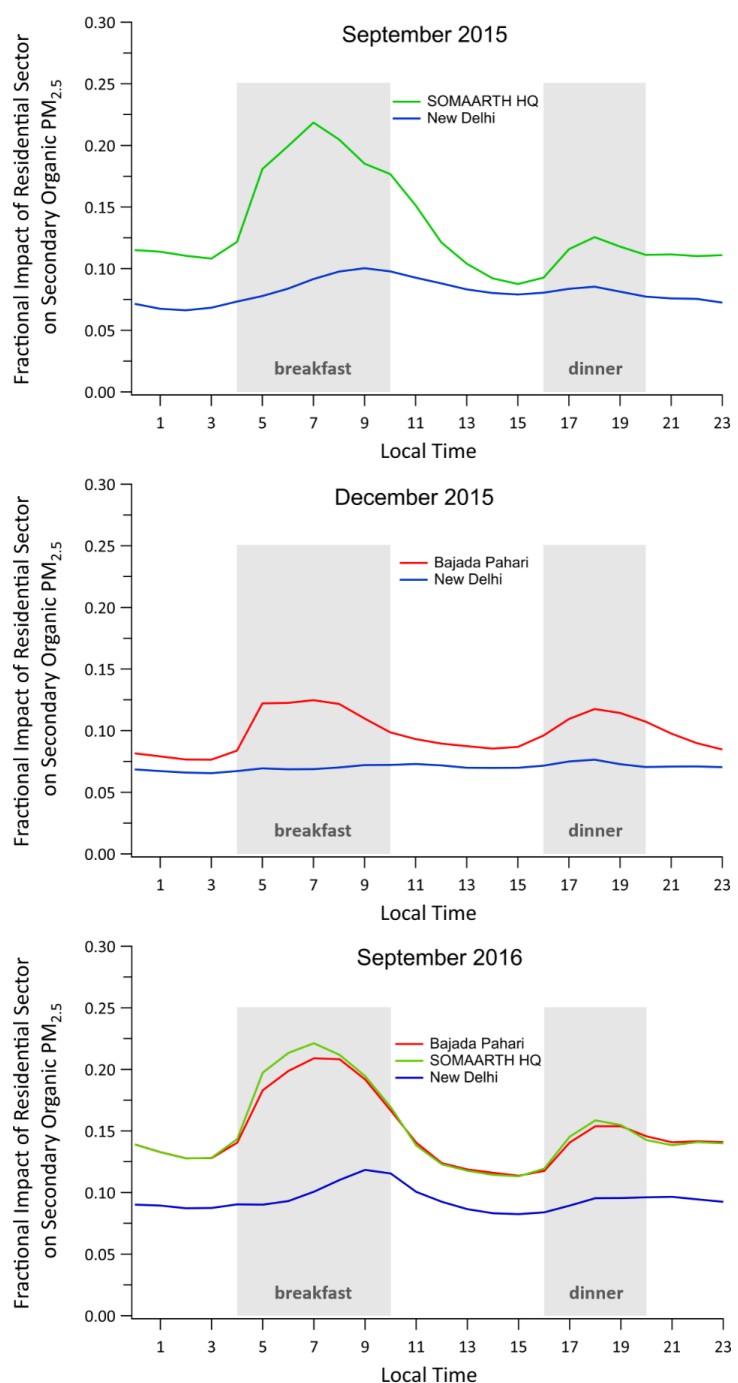

Figure 16.

