# Peer review of "Impacts of Household Sources on Air Pollution at Village and Regional Scales in India"

_Atmospheric Chemistry and Physics, 2018_

## Referee Comment (RC1) · Anonymous Referee #1 · 15 Jan 2019

General comments: The residential source is a major source sector for air pollution in India. Studying the impact of household source to air pollution is important for policy-making. The topic of this study is suitable for the ACP. However, major revisions are needed to further improve the study. The contents need to be re-organized, and the model performance should be evaluated cautiously before analyzing the results and drawing the conclusions. A discussion on the uncertainty and limitation of this study is also needed. Specific comments: 1 Line 10-15 on Page 6: "Thus, daily emission rates are generated for all species and sectors, except for the residential sector. All emissions are assumed to occur at Earth's surface. How do the authors derive the daily and hourly emission rates of different emission sectors for the CMAQ model? It seems limited information is available for the temporal profiles of most of the sectors other than

residential. And assuming all the emissions occur at Earth surface will overestimate the impact of high-stack emission sources to the ground-level concentrations, such as power and some industrial sectors. 2 Line 25-30 on page 10: why not directly run the GEOS-Chem model in 2015 and 2016 for the exact corresponding dates of CMAQ simulation? 3 Page 14-Page 15: One and a half pages are spent to review the previous studies on ozone simulation in India. The review is too long, which distracted the reader's attention. It's better to first present the author's results, and then have proper discussion with information from previous studies. Please re-organize the contents. 4 Line 20-25 on page 9: why do the authors choose Sep. 2015, Dec. 2015 and Sep 2016? Do the three periods has some relevance? What's the intention to choose three discrete periods? 5 Page 41: The comparison between simulation and observation shows that the model always underestimates the ozone concentration during night, close to zero. What is the reason? Is this reasonable? 6 Line 25-28 on page 15: "This overall comparison of predictions and observations would appear to be driven by the accuracy of the meteorological fields generated by the model. "Indeed, the performance of air pollutant predictions can be largely affected by the meteorological conditions simulated by WRF. It's necessary to evaluate the model performance of WRF prior to CMAQ evaluation, which may provide hints for the inconsistency between the observation and simulation. 7 Line 27-28: "In general, the degree of agreement between predicted and observed O3 levels in New Delhi over these periods should be considered as reasonable. "The conclusion here is not convincing. Please first do the statistics for the model evaluation, and then compare with previous studies to see whether it's reasonable or not. 8 Line 15-28 on page 15: please explain the reasons of underestimation and over estimation of ozone for different monitoring sites. 9 Page 16-Page 17: Somehow, the model performance for PM2.5 is also not satisfactory: large underestimation occurred in SOMAARTH HQ in Sep. 2015, Bajada Pahari in Dec. 2015, whereas over estimation occurred in Bajada Pahari in Sep. 2016, and New Delhi in Sep 2015 and Sep 2016. Before moving further, the authors should evaluate the model performance in details, analyze the reasons for the underestimation and

overestimation, and compare the model performance with other studies. 10 Line 2-4 on Page 18: "September 2015 and 2016, household energy-use activities account for up to 33% of ambient PM2.5 at SOMAARTH HQ and up to 28% at Bajada Pahari in September 2016."With current model performance, it's hard to trust the reliability of the results. Since a large proportion of PM2.5 mass concentration is missing compared with the observation. 11 Now that the study also analyzed the household contribution to SOA, a model evaluation for SOA is also needed. 12 A discussion on the uncertainty and limitation of this study is missing.
* * *

---

## Referee Comment (RC2) · Anonymous Referee #2 · 24 Jan 2019

**General Comments:** Rooney and coauthors have undertaken a worthwhile modeling study on the impact of residential combustion sources on air quality in rural and urban India. The goal, scope, and methods of the paper are well-suited to ACP. They should also be commended for their efforts processing emissions, meteorology and land use data for this simulation – all challenging tasks. The conclusions drawn are a good start to a conversation about contributions from residential burning to PM2.5 and ozone. However, I find significant areas of improvement possible in the explanation of methods, consideration of underlying assumptions, and discussion of results. Please address the concerns below.

**1. Quantitative and statistical analysis.** The manuscript includes many statements like "whereas for the end of the month, predicted levels match closely those observed"

[Figure]

(page 15, line 23), "In general, the degree of agreement between predicted and observed O3 levels in New Delhi over these periods should be considered as reasonable" (page 15, line 27-28) and "the finer resolution computations generally predict somewhat higher PM2.5 concentrations than he coarser computations" (page 16, lines 9-11). These kinds of statements need to be supported with rigorous statistical analysis. I would expect some subset of bias, error, RMSE, fractional bias, fractional error, correlation coefficient, index of agreement, etc. to be provided in any contemporary air modeling study with access to observation data. Aggregate metrics can often fail us if we rely too much on them, but they do help to summarize the overall performance of the model against other studies.

The authors also rely primarily on timeseries plots comparing model predictions to observations for raw concentrations. It is usually useful to look at these comparisons with diurnal plots as well as raw timeseries to see if there are persistent issues at certain times of day; I recommend adding these for the data presented in figures 6-11. It's also important to add error bars to illustrate variability on all diurnal timeseries plots, including figures 12 and 16.

I also would have expected to see maps of output data for metrics like fractional SOA contribution or fractional residential combustion contribution. Of course, it is just a model result, but I would think experts in air quality issues in India would find such maps interesting to ponder since this data cannot be obtained with existing measurements alone.

**2. Species-level evaluation.** Are there any observed speciation data that can help evaluate aspects of the model like the POA/SOA split? For example, OC/EC ratios might be instructive. It would also be useful to know if the model is predicting individual inorganic ion components well; do these data exist?

Unfortunately, there are no species-level data in this region for this period. We have now included discussion of model predictions of PM$_{2.5}$ speciation, see pages 10-11 and Table 10.

**3. Meteorology evaluation.** An evaluation of the meteorology fields has been completely omitted. This must be provided (e.g. supplemental information) or referenced

if published somewhere else. The met evaluation would also enhance the value of the timeseries plots of raw concentration data. If performance trends are seen across the month, it is important to find out if they are correlated with biases in any meteorological parameters.

**4. Model spinup.** The lack of model spinup for the individual scenarios is potentially troubling, considering that the authors are focusing on periods with meteorological conditions that favor accumulation (i.e. low wind speeds, low boundary layers, cold temperatures, etc.). Spin up is happening at the GEOS-Chem domain, so it is not as bad as starting from clean conditions, but there is a resolution change in going from the GEOS-CHem field to the CMAQ parent field and again for the nested field. It would be safer to at least consider omitting some of the timeseries for model spin-up. I recommend one of the following: a) rerun WRF and CMAQ simulations beginning about 10 days before each time period (i.e. August 20 for the Septemeber simulations). For emissions, if you don't have specific emissions for the spinup days, it should be safe to reuse appropriate days from the main simulation. For example, if August 20 is a weekday, make sure you pick a weekday from the main simulation. b) alternatively, cut off the first 7-10 days of each month form the existing results. Based on the timeseries data, it looks like making this revision will sacrifice minimal model-obs pairs for the PM dataset anyway.

**5. Impact of Conclusions.** Finally, the authors have summarized in the abstract and conclusions sections some important results of their work: a) residential combustion makes a significant contribution to air quality impacts in India, b) the model sometimes performs well, but sometimes does not, and this is a result of meteorological and emissions errors, and c) SOA makes up a significant but minor fraction of the total OA, with varying temporal profile between urban and rural areas. The authors have assembled a wealth of data and numerical predictive techniques to create this dataset; are there any more conclusions that can be drawn from this work if analyzed deeper?

For example, if one population-weights the emission rates of total PM or SOA precursors across the parent domain, what sources stand out as the most impactful? The authors have shown that residential emissions do not matter as much in New Delhi as they do at the rural sites, but how does this translate to people impacted? And what does the emissions dataset say are the most important sources of primary PM2.5, secondary PM2.5 precursors, and Ozone precursors?

Another question to consider (aspects of this are repeated in some specific issues below) is how confident are the authors in their estimate of SOA contributions given the uncertainty in emissions speciation of SVOCs and IVOCs, SOA yields of aromatics, and potential degradation pathways like homogeneous fragmentation or particle-phase reaction/photolysis, among other uncertainties? I'm not calling for a suite of sensitivity studies here, but considering the author list contains unique experts in emissions and chemistry, it would be great to understand and document where this team thinks most of the uncertainty lies at this point for SOA predictions in India. Or, alternatively, what parameters need more investigation before such conclusions can be drawn confidently?

**Specific Issues and Typos:**

1. Page 1: consider mentioning residential heating as well as cooking since it does become a significant part of the discussion in the results section.

2. Page 2, line 4: consider replacing "pollution species" with "pollutants" since I wouldn't consider PM2.5 to be a species, technically.

3. Page 3, Introduction: There's little to no mention of ozone here, which seems strange given its importance in the discussion later. Consider highlighting it here in parallel to PM.

4. Page 4, line 16: What does this sentence mean? There is no "prediction" for SOA in India? Do you mean there is no inferred measurement of it? Or that there is a dearth of reported model predictions to compare to? Please be more precise.

5. Page 7, line 14: This statement asserts that there are no emissions from heating in the study, but Figure 2 clearly shows parameters used to inform the heating emissions. Please explain the heating emissions more clearly.

6. Page 7, line 15: This sentence makes it sound like there are no solvent emissions because the simulation is in August. I suggest rewriting and stating why there are not any solvent emissions. I assume it's from a lack of input data?

7. Page 8, line 5: Burning of biomass fuel can give compounds with quite high OM:OC. How robust are your results to this simplifying assumption?

8. Page 8, line 6: PNCOM, PMC and PH2O are ignored in the emissions. Do you have references for the validity of assuming they are negligible? I understand omitting coarse PM since you are studying PM2.5. However, studies from airsheds like LA have shown that how a model treats inorganic aerosol in the coarse mode can be important for fine mode predictions. What is your basis for assuming negligible effect?

9. Page 8, line 9-10: How much of the PM2.5 emissions is carbon? Shouldn't it be included in this equation? Is it 100

10. CMAQv5.2 assumes one volatility distribution for all POA, while this study considers POA from a variety of fuels and burning methods. What uncertainties are introduced here and what impact do they have on conclusions?

11. The resolutions of the coarse and fine grids should be mentioned in the emissions methods discussion as it would help the reader understand right away the challenge involved in specifying emissions for this simulation.

12. Page 11, line 9: Did you use 4-D data assimilation to 'nudge' the met model to observations within the domain? If not, how often did you reinitialize the met model with realistic fields? One of these approaches should probably have been used to keep the met model realistic over a month-long simulation. If neither were used, it makes a met evaluation all the more important to demonstrate or reference to better understand

any growing biases.

13. Page 13: Could the authors please discuss why they feel it is appropriate to omit the pcSOA species, which is based on the SIMPLE method of Hayes et al. (2015)? Is this because the speciation studies used for this residential emissions are believed to account for all of the particle and vapor phase mass that may be missed in inventories like the United States NEI? Page 13, line 11 says SVOCs could not be quantified. This mass is potentially important and represents part of what the pcSOA is designed to address. Can the authors be more explicit about what classes of organic compounds are accounted for by their emissions speciation and which might be underrepresented? Discussing this in the context of volatility would suffice. For example, are IVOC precursors for SOA missing as well?

14. Page 14: Much of the content in the first four paragraphs of section "ozone" is better-suited for the introduction section than the results.

15. Page 16, line 12: Recommend replacing "closeness" with "similarity". How did you verify this statement? Perhaps make a timeseries of the emission rate of residential primary PM (median and error bars) across the child domain for the 4 km and 1 km cases?

16. Page 16, lines 17-20: Is this statement true at all sites? Or is there variation in the importance of daily-varying emissions like agricultural burning?

17. Page 18, lines 19-21: This sentence confused me. It says the emission inventories are the same, but that the main differences between scenarios is explained by differences in POA? I'm probably not understanding what scenarios the authors are comparing (e.g. seasons or resolution or sensitivities). It would be helpful to be more precise here.

18. Page 19, lines 11-18: I'm confused again. Which conclusions are from Fleming et al., and which are new to this study? If they're all from Fleming et al., how do they

compare with the contribution CMAQ is predicting for this work? Again, if aromatics are currently the dominant source of SOA, how much could be coming from IVOCs and potentially being missed?

19. Figure 1: 4 km grid cell should read 4 km grid domain.

20. Figure 4: Recommend adding some visual cues for the flow of information from the parent domain all the way through to the nested domain. For example, add "4 km resolution" to the "Gridded Airborne Concentrations" box. Then duplicate that box, but with "1 km resolution" instead. You could then color the arrows based on whether they are relevant for the 1 km or 4 km simulation. For example, purple for 4 km and orange for 1 km. So the arrow from GEOS-Chem to IC-BCs would be purple. And the arrow labeled "Nested Domain" would be orange. The arrows connecting IC-BCs, Emissions and Met to CMAQ would be both orange and purple. Also, shouldn't there be an arrow from GEOS-Chem to the Met box? And from the Met box to Emissions? Maybe this is more detail than you want to include? You probably don't want to use the phrase "Parent Domain" in the GEOS-Chem box since I assume this was a global simulation? CMAQv5.2 Chemical Transport "Module" should read "Model".

---

## Referee Comment (RC3) · Anonymous Referee #3 · 28 Jan 2019

This manuscript presents a multi-resolution modeling study of ozone and particulate matter in India, with a focus on the impact of residential combustion on ozone and particulate matter concentrations. The authors apply the CMAQ model at several different horizontal grid resolutions for several different time periods in 2015/2016. They then analyze the total concentrations of ozone/PM2.5 as compared to observations and examine the contribution of residential combustion of sources for cooking to the total PM2.5 and total SOA. Overall, the manuscript is generally well written. However, I agree with the points made by the other anonymous referees, and concur that the manuscript would benefit greatly from some major revisions. I will not repeat the general suggestions made by the other referees here, only to say that I agree with their major points. I will however provide some specific comments in addition to those made

by the other reviewers.

General Comments: Very briefly, I agree that the authors spend a lot of time discussion previous work upfront. It would be nice more details of their own analysis could be provided and then compared to the previous studies as appropriate. More details and analysis of both the WRF and CMAQ simulations is required, particularly the WRF simulations. This point was made by the other referees, but I think it is necessary to reiterate here the importance of providing an analysis of the performance of the WRF simulations. Hopefully there are observations available to provide such an analysis. Giving the reader a clearer picture of the strengths and weaknesses in the WRF simulations would be very beneficial to supporting the analysis of the CMAQ simulations and the conclusions the authors are attempting to make. Related, there are large biases in the CMAQ simulations that at bring into question some of the conclusions made by the authors, particularly those attributing percent contribution of residential cooking to total PM2.5 and total SOA. Providing a clearly picture of the CMAQ model performance would help support these conclusions (I refer the authors to the comments provided by the other referees). The authors should provide summary statistics (e.g. RMSE, bias, correlation) of the various simulations.

Specific Comments: Page 7, line 30: How different are the profiles for wood and dung? Why not use a combination of the two if the percent contribution in a grid cell is known?

Page 8, line 21: This appears to be sentence fragment to me. Please fix.

Page 10, line 26: As mentioned the other referee, it's a strange setup to be using the same month/day of the GEOS-Chem simulation to support the various CMAQ simulations. Why was this done?

Page 11: What input data did the authors use to drive the WRF simulations? Reanalysis data (if so, what was the horizontal resolution)? Were the same data used for all three horizontal domains? How many vertical layers were used in the WRF simulations, and what was the height of the lowest layer? What Cumulus Parameterization

(CP) scheme was used? Was a CP scheme applied for all the simulations?

Page 11, line 18: What version of MCIP was used? Also, please provide a relevant reference for MCIP.

Page 11: What was the CMAQ configuration (e.g. vertical layers, mixing scheme, etc.)? As mentioned by the other referee, it does not appear that a spin-up was applied. I agree with the referee that it would be beneficial to provide a spin-up period, even if not starting from profile ICs.

Page 15, line 25: As mentioned by the other referees, the author's statement that the "overall comparison of predictions and observations would appear to be driven by the accuracy of the meteorological fields" is on point, however no analysis of the accuracy of those fields is provided, so the reader has no idea what role the meteorological fields are actually playing in the analysis. So, these comparisons need to be provided since they are no doubt very important to the analysis and conclusions the authors are making.

Page 16, line 13: The statement that "The closeness of the 4 km and 1 km simulations reflects the closeness of the respective inventories" seems to only capture part of the story. What about differences in the meteorological performance between the two simulations? It would seem that the reader is intended to assume that the meteorological fields are also very similar between the 4km and 1km simulations. Is that really the case?

Page 19, lines 5-20: I'm confused whether the authors are describing new results from their own analysis or simply summering the results of Fleming et al.? Please clarify whether the statements are new results or simply summary of results from Fleming.

Page 20, line 10: The statement that "overall good agreement between observed and predicted levels O3 levels" is a bit generous. Based solely on the time-series plots and without any summary statistics, I think the authors at best could call the results reasonable. There are certainly frequently very large biases that make calling the results "good" difficult to support. However, I believe that even reasonable results are sufficient to support the conclusions the authors are making regarding ozone performance.

Page 21, line 5: The statement regarding the importance of replacing household combustion devices with modern technology is not strongly supported by the analysis presented. If anything, it would seem that reducing the impacts from agriculture burning would go much further in improving people's exposure to PM2.5 in this region. That is not to say that improving technology is not beneficial, it's just not the message I think that is delivered by the analysis as it is currently presented.

---

## Author Comment (AC1) · 9 Apr 2019

**B. Rooney et al., "Impacts of Household Sources on Air Pollution at Village and Regional Scales in India"**

**Responses to Referee Comments**

We sincerely appreciate the care with which the Reviewers evaluated this manuscript. This was a particularly challenging project, combining gaseous and particulate emissions inventories over India, WRF meteorological modeling, gas- and aerosol-phase chemical transport modeling by GEOS-Chem and CMAQ, and analysis of in situ measurements in north-central India. (The paper includes 21 authors from 7 institutions in the United States and India.) In the following responses, we address each of the comments of the Reviewers. All the simulations in the work have been redone. This includes comprehensive meteorology with WRF, domain boundary conditions with GEOS-Chem, and gas-phase and particulate matter concentrations with CMAQ. GEOS-Chem was run for each simulation time period (September 2015, December 2015, and September 2016) to supply the boundary conditions to CMAQ. The meteorology input was re-generated by WRF based on more accurate parameterizations and the most up-to-date initialization data. We now include more detailed statistical analyses of predictions and observations than in the original manuscript. Use of updated meteorology resulted in considerably improved agreement between predictions and observations for all PM components. As a result, a major fraction of the paper has been rewritten in accord with the revised computational simulations. In the accompanying revised manuscript, all new material is presented in red typeface.

**Referee 1**
*General comments:*
The residential source is a major source sector for air pollution in India. Studying the impact of household source to air pollution is important for policymaking. The topic of this study is suitable for the ACP. However, major revisions are needed to further improve the study. The contents need to be re-organized, and the model performance should be evaluated cautiously before analyzing the results and drawing the conclusions. A discussion on the uncertainty and limitation of this study is also needed.

*Specific comments:*
1. Line 10-15 on page 6: "Thus, daily emission rates are generated for all species and sectors, except for the residential sector. All emissions are assumed to occur at Earth's surface." How do the authors derive the daily and hourly emission rates of different emission sectors for the CMAQ model? It seems limited information is available for the temporal profiles of most of the sectors other than residential. And assuming all the emissions occur at Earth surface will overestimate the impact of high-stack emissions sources to the ground-level concentrations, such as power and some industrial sectors.

We have replaced the previous material with a new description, see page 5, lines 5 through line 31.

2. Line 25-30 on page 10: Why not directly run the GEOS-Chem model in 2015 and 2016 for the exact corresponding dates of CMAQ simulation?

We have now run GEOS-Chem directly for September 2015, December 2015, and September 2016. All CMAQ simulations have been re-run with the appropriate GEOS-Chem boundary conditions.

3. Page 14-15: One and a half pages are spent to review the previous studies on ozone simulation in India. The review is too long, which distracted the reader's attention. It's better to first present the author's results, and then have proper discussion with information from previous studies. Please re-organize the contents.

We have shortened the review of previous studies on $O_3$ in India. Lines 7-22 on p. 14 of the original manuscript have been removed.

4. Line 20-25 on page 9: Why do the authors choose Sep. 2015, Dec. 2015 and Sep. 2016? Do the three periods have some relevance? What's the intention to choose three discrete periods?

The periods September 2015, December 2015, and September 2016 were chosen because actual measurements ($PM_{2.5}$, $O_3$, temperature, and wind direction) were carried out in the region of interest during these months.

5. Page 41: The comparison between simulation and observation shows that the model always underestimates the ozone concentration during night, close to zero. What is the reason? Is this reasonable?

At night, when radiation is minimal, $O_3$ is consumed by NO titration reactions. In the present study, CMAQ simulations overpredicted NO, particularly at night when NO peaks, leading to the underestimation of ozone at night. This effect could be exacerbated by an underestimated planetary boundary layer height or weak vertical mixing.

6. Line 25-28 on page 15: "This overall comparison of predictions and observations would appear to be driven by the accuracy of the meteorological fields generated by the model." Indeed, the performance of air pollutant predictions can be largely affected by the meteorological conditions simulated by WRF. It's necessary to evaluate the model performance of WRF prior to CMAQ evaluation, which may provide hints for the inconsistency between the observation and simulation.

We now have added a new figure and a new table in the text to show the evaluation of WRF simulated meteorology against the available surface observations at different sites during the same periods. The new Figure 5 shows that there is generally good agreement of surface temperature between WRF and observations for all three months. The surface wind direction is also found consistent between model and observations for each site and each month (Table 9). The simulated near-surface wind speeds are overestimated by WRF, with an averaged mean-bias (MB) of about +1.5 m/s. Such a bias is partly a result of the difference in the definition of "near surface" between the model and observations. The new Figure 5 and new Table 9 follow.

[Figure]

Figure 5. Evaluation of WRF simulated surface temperature versus ground observations.

| | | Bajada Pahari | | | SOMAARTH HQ | | | West New Delhi | | | South New Delhi | | |
|---|---|---|---|---|---|---|---|---|---|---|---|---|---|
| | | Sep '15 | Dec '15 | Sep '16 | Sep '15 | Dec '15 | Sep '16 | Sep '15 | Dec '15 | Sep '16 | Sep '15 | Dec '15 | Sep '16 |
| **Temperature (°C)** | PRE | - | 15.28 (4.59) | 30.10 (3.19) | 29.27 (3.48) | - | 30.22 (3.06) | 30.45 (3.79) | 16.59 (4.91) | 30.07 (3.05) | 30.32 (3.74) | 17.59 (4.82) | 29.96 (30.3) |
| | OBS | - | 15.62 (4.91) | 30.86 (5.67) | 32.15 (4.12) | - | 33.26 (5.31) | 32.80 (3.60) | 19.04 (3.66) | 31.46 (2.33) | 28.48 (4.30) | 12.58 (5.52) | 29.22 (4.22) |
| | MB | - | -0.34 | -0.76 | -2.89 | - | -3.04 | -2.35 | -2.45 | -1.38 | 1.84 | 5.02 | 0.74 |
| | ME | - | 1.60 | 3.08 | 2.92 | - | 3.07 | 3.03 | 2.58 | 1.54 | 2.11 | 5.02 | 2.37 |
| | RMSE | - | 2.20 | 3.71 | 3.39 | - | 3.99 | 3.58 | 2.99 | 1.88 | 2.50 | 5.33 | 2.75 |
| **Wind Speed (m·s$^{-1}$)** | PRE | - | 2.91 (1.17) | 2.31 (1.07) | - | - | 2.01 (0.66) | - | - | 2.57 (1.28) | 2.80 (1.27) | 2.72 (1.08) | 2.74 (1.39) |
| | OBS | - | 1.18 (0.75) | 0.73 (0.40) | - | - | 0.55 (0.30) | - | - | 1.03 (0.51) | 1.26 (0.83) | 0.94 (0.71) | 1.18 (0.79) |
| | MB | - | 1.72 | 1.58 | - | - | 1.46 | - | - | 1.54 | 1.54 | 1.77 | 1.56 |
| | ME | - | 1.75 | 1.62 | - | - | 1.50 | - | - | 1.58 | 1.61 | 1.82 | 1.62 |
| | RMSE | - | 1.96 | 1.85 | - | - | 1.66 | - | - | 1.88 | 1.85 | 2.01 | 1.84 |
| **Wind Direction (°)** | PRE | - | 247 (111) | 116 (45) | 272 (70) | - | 111 (51) | - | - | 179 (98) | 206 (118) | 254 (97) | 191 (96) |
| | OBS | - | 259 (57) | 102 (41) | 255 (58) | - | 110 (48) | - | - | 181 (97) | 198 (45) | 224 (44) | 228 (50) |
| | MB | - | 0.14 | 14 | 16 | - | -0.14 | - | - | -6 | 9 | 35 | -34 |
| | ME | - | 51 | 38 | 44 | - | 32.71 | - | - | 49 | 94 | 74 | 75 |
| | RMSE | - | 66 | 51 | 64 | - | 47.50 | - | - | 64 | 106 | 87 | 90 |

Table 9. Quantification of WRF model biases in meteorological fields. PRE is mean predictions; OBS is mean observations; MB is mean bias; ME is mean error; and RMSE is root mean square error. Standard deviation of predictions and observations are noted in parentheses.

7. Line 27-28 [on page 15]: "In general, the degree of agreement between predicted and observed O$_3$ levels in New Delhi over these periods should be considered as reasonable." The conclusion here is not convincing. Please first do the statistics for the model evaluation, and then compare with previous studies to see whether it's reasonable or not.

We have now redone all CMAQ simulations. We address the agreement between the new O$_3$ predictions and observations in New Delhi on page 12. Statistics are included in Table 10.

8.  Line 15-28 on page 15:  Please explain the reasons of underestimation and over estimation of ozone for different monitoring sites.

The results of ozone simulations in the present study are generally consistent with those of previous simulations over India. Virtually all studies of ozone modeling in urban areas in the literature show sites where concentrations are overestimated and sites where concentrations are underestimated. For example, also using WRF-CMAQ, Kota et al. (2018) showed that the relative bias in ozone simulation ranges from −30% to +50% in major cities of India. In South New Delhi, for example, the bias in $O_3$ predictions in the present study lies between -2.67 and +7.01 µg m$^{-3}$, as compared to the observations of 29.28 to 62.76 µg m$^{-3}$ (see Table 10).

Kota, S.H., Guo, H., Myllyvirta, L., Hu, J., Sahu, S., Garaga, R., Ying, Q., Gao, A., Dahiya, S., Wang, Y., and Zhang, H., 2018. Year-long simulation of gaseous and particulate air pollutants in India, Atmospheric Environment, 180, 244-255.

9.  Page 16-17:  Somehow, the model performance for PM$_{2.5}$ is also not satisfactory:  large underestimation occurred in SOMAARTH HQ in Sep. 2015, Bajada Pahari in Dec. 2015, whereas over estimation occurred in Bajada Pahari in Sep. 2016, and New Delhi in Sep. 2015 and Sep. 2016.  Before moving further, the authors should evaluate the model performance in details, analyze the reasons for the underestimation and overestimation, and compare the model performance with other studies.

All CMAQ simulations have been redone with updated meteorology from WRF (see response to Referee 1, Comment 6), updated GEOS-Chem boundary conditions (see response to Referee 1, Comment 2), and updated emissions inventory to include biogenic VOC emissions (see response to Referee 1, Comment 1). The comparisons of predictions and observations are now entirely revised and with new Figures 6 through 9. A discussion of model performance is provided on pages 15-16 and statistics are detailed in Table 10.

10.  Line 2-4 on page 18:  "September 2015 and 2016, household energy-use activities account for up to 33% of ambient PM$_{2.5}$ at SOMAARTH HQ and up to 28% at Bajada Pahari in September 2016."  With current model performance, it's hard to trust the reliability of the results. Since a large proportion of PM$_{2.5}$ mass concentration is missing compared with the observation.

All CMAQ simulations have been redone (see response to Referee 1, Comment 9). Statistical comparisons of predictions and observations are now included, see Table 10. New discussion of the contribution from household energy-use activities is provided on page 11, line 22 through page 12, line 2.

11.  Now that the study also analyzed the household contribution to SOA, a model evaluation for SOA is also needed.

Direct measurements of the SOA fraction of PM are, unfortunately, not available in this region. Nonetheless, information is given throughout the manuscript on the levels of SOA predicted.

12. A discussion on the uncertainty and limitation of this study is missing.

A discussion of uncertainty is now provided, see page 13, lines 31-36.

**Referee 2**
*General comments:*
Rooney and coauthors have undertaken a worthwhile modeling study on the impact of residential combustion sources on air quality in rural and urban India. The goal, scope, and methods of the paper are well-suited to ACP. They should also be commended for their efforts processing emissions, meteorology and land use data for this simulation – all challenging tasks. The conclusions drawn are a good start to a conversation about contributions from residential burning to $PM_{2.5}$ and ozone. However, I find significant areas of improvement possible in the explanation of methods, consideration of underlying assumptions, and discussion of results. Please address the concerns below.

*1. Quantitative and statistical analysis.* The manuscript includes many statements like "whereas for the end of the month, predicted levels match closely those observed" (page 15, line 23), "In general, the degree of agreement between predicted and observed $O_3$ levels in New Delhi over these periods should be considered as reasonable" (page 15, line 27-28) and "the finer resolution computations generally predict somewhat higher $PM_{2.5}$ concentrations than the coarser computations" (page 16, lines 9-11). These kinds of statements need to be supported with rigorous statistical analysis. I would expect some subset of bias, error, RMSE, fractional bias, fractional error, correlation coefficient, index of agreement, etc. to be provided in any contemporary air modeling study with access to observation data. Aggregate metrics can often fail us if we rely too much on them, but they do help summarize the overall performance of the model against other studies.

The authors also rely primarily on timeseries plots comparing model predictions to observations for raw concentrations. It is usually useful to look at these comparisons with diurnal plots as well as raw timeseries to see if there are persistent issues at certain times of day; I recommend adding these for the data presented in figures 6-11. It's also important to add error bars to illustrate variability on all diurnal timeseries plots, including figures 12 and 16.

I also would have expected to see maps of output data for metrics like fractional SOA contribution or fractional residential combustion contribution. Of course, it is just a model result, but I would think experts in air quality issues in India would find such maps interesting to ponder since this data cannot be obtained with existing measurements alone.

Statistical analyses of predictions and observations are now described in detail, see Table 10. Timeseries plots comparing model predictions to observations have been completely revised to include updated model predictions and diurnal plots.

*2. Species-level evaluation.* Are there any observed speciation data that can help evaluate aspects of the model like the POA/SOA split? For example, OC/EC ratios might be instructive. It would also be useful to know if the model is predicting individual inorganic ion components well; do these data exist?

Unfortunately, there are no species-level data in this region for this period. We have now included discussion of model predictions of $PM_{2.5}$ speciation, see pages 10-11 and Table 10.

**3. _Meteorology evaluation._**  An evaluation of the meteorology fields has been completely omitted.  This must be provided (e.g. supplemental information) or referenced if published somewhere else.  The met evaluation would also enhance the value of the timeseries plots of raw concentration data.  If performance trends are seen across the month, it is important to find out if they are correlated with biases in any meteorological parameters.

See response to Referee 1, Comment 6.

**4. _Model spinup._**  This lack of model spinup for the individual scenarios is potentially troubling, considering that the authors are focusing on periods with meteorological conditions that favor accumulation (i.e. low wind speeds, low boundary layers, cold temperatures, etc.).  Spin up is happening at the GEOS-Chem domain, so it is not as bad as starting from clean conditions, but there is a resolution change in going from the GEOS-Chem field to the CMAQ parent field and again for the nested field.  It would be safer to at least consider omitting some of the timeseries for model spin-up.  I recommend one of the following:  a) rerun WRF and CMAQ simulations beginning about 10 days before each time period (i.e. August 20 for September simulations).  For emissions, if you don't have specific emissions for the spinup days, it should be safe to reuse appropriate days from the main simulation.  For example, if August 20 is a weekday, make sure you pick a weekday from the main simulation.  b) alternatively, cut off the first 7-10 days of each month from the existing results.  Based on the timeseries data, it looks like making this revision will sacrifice minimal model-obs pairs for the PM dataset anyway.

All CMAQ simulations have been redone (see response to Referee 1, Comment 9), with slightly longer time periods. The September 2015 and September 2016 simulations now begin five days earlier. Additionally, the first five days of each simulation were excluded as spin-up from analysis and plots.

**5. _Impact of Conclusions._**  Finally, the authors have summarized in the abstract and conclusions sections some important results of their work:  a) residential combustion makes a significant contribution to air quality impacts in India, b) the model sometimes performs well, but sometimes does not, and this is a result of meteorological and emissions errors, and c) SOA makes up a significant but minor fraction of the total OA, with varying temporal profile between urban and rural areas.  The authors have assembled a wealth of data and numerical predictive techniques to create this dataset; are there any more conclusions that can be drawn from this work if analyzed deeper?

For example, if one population-weights the emission rates of total PM or SOA precursors across the parent domain, what sources stand out as the most impactful?  The authors have shown that residential emissions do not matter as much in New Delhi as they do at the rural sites, but how does this translate to people impacted?  And what does the emissions dataset say are the most important sources of primary $PM_{2.5}$, secondary $PM_{2.5}$ precursors, and Ozone precursors?

Another question to consider (aspects of this are repeated in some specific issues below) is how confident are the authors in their estimate of SOA contributions given the uncertainty in emissions speciation of SVOCs and IVOCs, SOA yields of aromatics, and potential degradation pathways like homogeneous fragmentation or particle-phase reaction/photolysis, among other uncertainties? I'm not calling for a suite of sensitivity studies here, but considering the author list contains unique experts in emissions and chemistry, it would be great to understand and document where this team thinks most of the uncertainty lies at this point for SOA predictions in India. Or, alternatively, what parameters need more investigation before such conclusions can be drawn confidently?

*Specific Issues and Typos:*
1. Page 1: Consider mentioning residential heating as well as cooking since it does become a significant part of the discussion in the results section.

Residential heating is not in the inventory, see response to comment 5 below.

2. Page 2, line 4: Consider replacing "pollution species" with "pollutants" since I wouldn't consider $PM_{2.5}$ to be a species, technically.

Correction made.

3. Page 3, Introduction: There's little to no mention of ozone here, which seems strange given its importance in the discussion later. Consider highlighting it here in parallel to PM.

Now included.

4. Page 4, line 16: What does this sentence mean? There is no "prediction" for SOA in India? Do you mean there is no inferred measurement of it? Or that there is a dearth of reported model predictions to compare to? Please be more precise.

We have clarified this statement. Whereas the CMAQ model predicts the fraction of $PM_{2.5}$ that is SOA, ambient measurements do not delineate the fraction of SOA.

5. Page 7, line 14: This statement asserts that there are no emissions from heating in the study, but Figure 2 clearly shows parameters used to inform the heating emissions. Please explain the heating emissions more clearly.

Residential emission rates for $PM_{2.5}$, black carbon (BC), organic carbon (OC), CO, $NO_x$, $CH_4$, $CO_2$, and total non-methane hydrocarbons (NMHC) were generated from SPEW. While SPEW incorporates temperature-dependent heating combustion activity, the inventory assumes temperatures too high for this activity to take effect. Thus, our inventory has no emissions from heating. Nonetheless, Figure 2 shows the fraction of daily household emissions from all quantifiable energy-use activities.

6.  Page 7, line 15:  This sentence makes it sound like there are no solvent emissions because the simulation is in August.  I suggest rewriting and stating why there are not any solvent emissions.  I assume it's from a lack of input data?

This statement has now been clarified. Solvent emissions are not included in the simulations for lack of specific input data.

7.  Page 8, line 5:  Burning of biomass fuel can give compounds with quite high OM:OC.  How robust are your results to this simplifying assumption?

The issue is the accuracy with which the mass of biomass fuel emissions is simulated by CMAQ. Specific data on the OM:OC ratio of biomass burning emissions in this region are not available. CMAQ has been used in other studies to predict the mass of combustion emissions based on the correlations in CMAQ.  While no specific data are available to evaluate the extent to which the CMAQ algorithm accurately represents the concentration of biomass burning emissions, $PM_{2.5}$ was speciated using Jayarathne et al. (2018), who studied emissions from burning wood and from burning a mixture of wood and dung, but not purely dung. The PM emission profiles for these fuels are now given in Table 3.

8.  Page 8, line 6:  PNCOM, PMC and $PH_2O$ are ignored in the emissions.  Do you have references for the validity of assuming they are negligible?  I understand omitting coarse PM since you are studying $PM_{2.5}$.  However, studies from airsheds like LA have shown that how a model treats inorganic aerosol in the coarse mode can be important for fine mode predictions. What is your basis for assuming negligible effect?

Coarse PM is included in the emissions (see Tables 1-3), and this line has been revised accordingly. PNCOM and $PH_2O$ are omitted for lack of information.

9.  Page 8, line 9-10:  How much of the $PM_{2.5}$ emissions is carbon?  Shouldn't it be included in this equation?

The equation and lines have been corrected, see page 6, lines 28-30.

10. CMAQv5.2 assumes one volatility distribution for all POA, while this study considers POA from a variety of fuels and burning methods.  What uncertainties are introduced here and what impact do they have on conclusions?

No explicit information is available on the volatility of POA. However, it is likely that POA has a sufficiently low volatility, as assumed in CMAQ v5.2. It is not expected that this assumption introduces any significant uncertainty.

11.  The resolutions of the coarse and fine grids should be mentioned in the emissions methods discussion as it would help the reader understand right away the challenge involved in specifying emissions for this simulation.

The resolution is now given.

12. Page 11, line 9: Did you use 4-D data assimilation to 'nudge' the met model to observations within the domain? If not, how often did you reinitialize the met model with realistic fields? One of these approaches should probably have been used to keep the met model realistic over a month-long simulation. If neither were used, it makes a met evaluation all the more important to demonstrate or reference to better understand any growing biases.

We do not use any nudging or data simulation approach in our WRF simulations. Three one-month simulations were performed without any re-initialization, which is a typical way to run the WRF model for regional weather/climate simulations [*Wu et al.*, 2017; *Kota et al.*, 2018]. Our simulated meteorological fields also show no systematic drift-away within a month, and they are quite consistent with ground-based observations, as discussed in our response to the major comment 3 above.

Kota *et al.*, Year-long simulation of gaseous and particulate air pollutants in India, Atmospheric Environment, 180, 244-255 (2018).
Wu *et al.*, WRF-Chem simulation of aerosol seasonal variability in the San Joaquin Valley Atmos. Chem. Phys. 17, 7291-7309 (2017).

13. Page 13: Could the authors please discuss why they feel it is appropriate to omit the pcSOA species, which is based on the SIMPLE method of Hayes et al. (2015)? Is this because the speciation studies used for these residential emissions are believed to account for all of the particle and vapor phase mass that may be missed in inventories of the United States NEI? Page 13, line 11 says SVOCs could not be quantified. This mass is potentially important and represents part of what the pcSOA is designed to address. Can the authors be more explicit about what classes of organic compounds are accounted for by their emissions speciation and which might be underrepresented? Discussing this in the context of volatility would suffice. For example, are IVOC precursors for SOA missing as well?

We have now included pcSOA in our revised CMAQ simulations.

14. Page 14: Much of the content in the first four paragraphs of section "ozone" is better suited for the introduction section than the results.

As noted above, the discussion of ozone simulations has been significantly shortened.

15. Page 16, line 12: Recommend replacing "closeness" with "similarity". How did you verify this statement? Perhaps make a timeseries of the emission rate of residential primary PM (median and error bars) across the child domain for the 4 km and 1 km cases?

The 1 km simulations have been removed from the revised manuscript as the initial results showed little difference between the two resolutions. Additionally, non-residential emissions were adapted from an Indian inventory with a native resolution of 36 km and boundary conditions have a resolution of $2° \times 2.5°$. Because of the coarseness of these inputs, it is unlikely that the difference between 1 km and 4 km chemical transport simulations is significant.

16. Page 16, lines 17-20: Is this statement true at all sites? Or is there variation in the importance of daily-varying emissions like agricultural burning?

All CMAQ simulations have been updated with new inputs and the comparisons of predictions and observations are now entirely revised (see response to Referee 1, Comment 9).

17. Page 18, lines 19-21: This sentence confused me. It says the emission inventories are the same, but that the main differences between scenarios is explained by differences in POA? I'm probably not understanding what scenarios the authors are comparing (e.g. seasons or resolution or sensitivities). It would be helpful to be more precise here.

With the new simulations, this comment is no longer relevant.

18. Page 19, lines 11-18: I'm confused again. Which conclusions are from Fleming et al., and which are new to this study? If they're all from Fleming et al., how do they compare with the contribution CMAQ is predicting for this work? Again, if aromatics are currently the dominant source of SOA, how much could be coming from IVOCs and potentially being missed?

The summary of results from Fleming et al. has been removed.

19. Figure 1: 4 km grid cell should read 4 km grid domain.

Figure 1 has been revised.

20. Figure 4: Recommend adding some visual cues for the flow of information from the parent domain all the way through to the nested domain. For example, add "4 km resolution" to the "Gridded Airborne Concentrations" box. Then duplicate that box, but with "1 km resolution" instead. You could then color the arrows based on whether they are relevant for the 1 km or 4 km simulation. For example, purple for 4 km and orange for 1 km. So the arrow from GEOS-Chem to IC-BCs would be purple. And the arrow labeled "Nested Domain" would be orange. The arrows connecting IC-BCs, Emissions and Met to CMAQ would be both orange and purple. Also, shouldn't there be an arrow from GEOS-Chem to the Met box? And from the Met box to Emissions? Maybe this is more detail than you want to include? You probably don't want to use the phrase "Parent Domain" in the GEOS-Chem box since I assume this was a global simulation? CMAQv5.2 Chemical Transport "Module" should read "Model".

With the omission of the nested 1 km simulations, we have removed Figure 4.

**Referee 3**
This manuscript presents a multi-resolution modeling study of ozone and particulate matter in India, with a focus on the impact of residential combustion on ozone and particulate matter concentrations. The authors apply the CMAQ model at several different horizontal grid resolutions for several different time periods in 2015/2016. They then analyze the total

concentrations of ozone/PM$_{2.5}$ as compared to observations and examine the contribution of residential combustion of sources for cooking to the total PM$_{2.5}$ and total SOA. Overall, the manuscript is generally well written. However, I agree with the points made by the other anonymous referees, and concur that the manuscript would benefit greatly from some major revisions. I will not repeat the general suggestions made by the other referees here, only to say that I agree with their major points. I will however provide some specific comments in addition to those made by the other reviewers.

*General Comments:*
Very briefly, I agree that the authors spend a lot of time discussion previous work upfront. It would be nice more details of their own analysis could be provided and then compared to the previous studies as appropriate. More details and analysis of both the WRF and CMAQ simulations is required, particularly the WRF simulations. The point was made by the other referees, but I think it is necessary to reiterate here the importance of providing an analysis of the performance of the WRF simulations. Hopefully there are observations available to provide such an analysis. Giving the reader a clearer picture of the strengths and weaknesses in the WRF simulations would be very beneficial to supporting the analysis of the CMAQ simulations and the conclusions the authors are attempting to make. Related, there are large biases in the CMAQ simulations that at bring into question some of the conclusions made by the authors, particularly those attributing percent contribution of residential cooking to total PM$_{2.5}$ and total SOA. Providing a clearly picture of the CMAQ model performance would help support these conclusions (I refer the authors to the comments provided by the other referees). The authors should provide summary statistics (e.g. RMSE, bias, correlation) of the various simulations.

See responses above to comments of Referees 1 and 2.

*Specific comments:*
Page 7, line 30: How different are the profiles for wood and dung? Why not use a combination of the two if the percent contribution in a grid cell is known?

PM$_{2.5}$ was speciated using Jayarathne et al. (2018), who studied emissions from burning wood and from burning a mixture of wood and dung, but not purely dung. The PM emission profiles for these fuels are now given in Table 3.

Page 8, line 21: This appears to be sentence fragment to me. Please fix.

The sentence in question is no longer in the paper.

Page 10, line 26: As mentioned the other referee, it's a strange setup to be using the same month/day of the GEOS-Chem simulation to support the various CMAQ simulations. Why was this done?

This has been addressed. See response to Referee 1.

Page 11: What input data did the authors use to drive the WRF simulations? Reanalysis data (if so, what was the horizontal resolution)? Were the same data used for all three horizontal domains? How many vertical layers were used in the WRF simulations, and what was the height of the lowest layer? What Cumulus Parameterization (CP) scheme was used? Was a CP scheme applied for all the simulations?

Now we update our meteorological forcing by switching to the latest version of ECMWF-ERA5 which was just released in January 2019. These reanalysis data are on a 30 km grid and resolve the atmosphere using 137 levels from the surface up to a height of 80 km. In the WRF model, we use 24 levels in vertical, consistent with our setup in the CMAQ model. The lowest layer is about 50 m. No cumulus parameterization is turned on in our simulation, as we only have one domain with 4 km resolution.

Page 11, line 18: What version of MCIP was used? Also, please provide a relevant reference for MCIP.

MCIPv4.4 (Otte et al., 2010) was used to process WRF for CMAQ.

Otte, T. L. and Pleim, J. E.: The Meteorology-Chemistry Interface Processor (MCIP) for the CMAQ modeling system: updates through MCIPv3.4.1, Geosci. Model Dev., 3, 243-256, https://doi.org/10.5194/gmd-3-243-2010, 2010.

Page 11: What was the CMAQ configuration (e.g. vertical layers, mixing scheme, etc.)? As mentioned by the other referee, it does not appear that a spin-up was applied. I agree with the referee that it would be beneficial to provide a spin-up period, even if not starting from profile ICs.

Analysis of the updated CMAQ simulations excludes five days for spin-up (see response to Referee 2, Comment 4).

Page 15, line 25: As mentioned by the other referees, the author's statement that the "overall comparison of predictions and observations would appear to be driven by the accuracy of the meteorological fields" is on point, however no analysis of the accuracy of those fields is provided, so the reader has no idea what role the meteorological fields are actually playing in the analysis. So, these comparisons need to be provided since they are no doubt very important to the analysis and conclusions the authors are making.

See response to Referee 1, Comment 6.

Page 16, line 13: The statement that "The closeness of the 4 km and the1 km simulations reflects the closeness of the respective inventories" seems to only capture part of the story. What about differences in the meteorological performance between the two simulations? It would seem that the reader is intended to assume that the meteorological fields are also very similar between the 4 km and 1 km simulations. Is that really the case?

The 1 km simulations have been omitted from the revised manuscript (see Referee 2, Comment 15).

Page 19, lines 5-20: I'm confused whether the authors are describing new results from their own analysis or simply summarizing the results of Fleming et al.? Please clarify whether the statements are new results or simply summary of results from Fleming.

Discussion of results from Fleming et al. has been removed as not essential to the present work.

Page 20, line 10: The statement that "overall good agreement between observed and predicted levels $O_3$ levels" is a bit generous. Based solely on the time-series plots and without any summary statistics, I think the authors at best could call the results reasonable. There are certainly frequently very large biases that make calling the results "good" difficult to support. However, I believe that even reasonable results are sufficient to support the conclusions the authors are making regarding ozone performance.

Summary statistics are now provided in Table 10. See response to Referee 1, Comment 7.

Page 21, line 5: The statement regarding the importance of replacing household combustion devices with modern technology is not strongly supported by the analysis presented. If anything, it would seem that reducing the impacts from agriculture burning would go much further in improving people's exposure to $PM_{2.5}$ in this region. That is not to say that improving technology is not beneficial, it's just not the message I think that is delivered by the analysis as it is currently presented.

As noted above, all CMAQ simulations have been redone corresponding to more accurate meteorological fields. As a consequence, predictions of the average fractions of total anthropogenic $PM_{2.5}$ and secondary organic $PM_{2.5}$ due to household emissions increased (see Figure 12). In addition, the minimal impact of the residential sector on $PM_{2.5}$ remains consistently higher than the previous simulations. We feel it is appropriate to state that more modern cooking technologies would be beneficial. Therefore, we have left the sentence in question in the paper.

**New text as the result of revising the manuscript in response to the reviews appears in red.**

[revised manuscript text omitted]

15     where PM$_{Non\text{-}Residential}$ PM$_{Total}$ are the predictions of anthropogenic PM$_{2.5}$ (top) or secondary organic PM$_{2.5}$ (bottom) from the non-residential and total emission scenario, respectively, averaged over simulation durations (Table 7). Computations were carried out at 4 km resolution. Statistics are shown in Table 10.

Fig. 13. Predicted O$_3$ (left) and average diurnal cycle (right) for 12/20/15 – 12/31/15 (top), 09/07/09/30/15 (middle),
20     and 09/20/16 – 09/30/16 (bottom) in West New Delhi (pink), and South New Delhi (blue). Standard deviations of the diurnal profiles for observations and predictions are indicated, respectively, by colored shading. Diurnal profiles were averaged over simulation durations (Table 7). Computations were carried out at 4 km resolution.

[Figure]

**Figure 1.**

[Figure]

**Figure 2.**

[Figure]

**Figure 3.**

**CMAQv5.2 CB6R3 and AERO6 Anthropogenic SOA Treatment**

[Figure]

**Figure 4.**

[Figure]

**Figure 5.**

[Figure]

**Figure 6.**

**SOMAARTH HQ**

[Figure]

**Figure 7.**

[Figure]

**Figure 8.**

[Figure]

**Figure 9.**

[Figure]

**Figure 10.**

[Figure]

**Figure 11.**

[Figure]

**Figure 12.**

[Figure]

**Figure 13.**